# TITAM (v1.0): Time Independent Tracking Algorithm for Medicanes

Enrique Pravia-Sarabia[1], Juan José Gómez-Navarro[1], Pedro Jiménez-Guerrero[1,2], and Juan Pedro Montávez[1]

[1]Physics of the Earth, Regional Campus of International Excellence (CEIR) "Campus Mare Nostrum", University of Murcia, 30100 Murcia, Spain

[2]Biomedical Research Institute of Murcia (IMIB-Arrixaca), 30120 Murcia, Spain

**Correspondence:** Juan Pedro Montávez (montavez@um.es)

**Abstract.**

This work aims at presenting TITAM, a time independent tracking algorithm specifically suited for medicanes. In the last decades, the study of medicanes has been repeatedly addressed given their potential to damage coastal zones. Their hazardous associated meteorological conditions have converted them in a major threat. Even though the medicanes similarities with
5 tropical cyclones have been widely studied in terms of genesis mechanisms and structure, the fact that the former ones appear in baroclinic environments, as well as the limited extension of the Mediterranean basin, make them prone to maintain their warm-cored and symmetric structure for short time periods. Thus, the usage of a measure for the warm-core nature of the cyclone, namely the Hart conditions, stands as a key factor for a successful identification of the medicane. Furthermore, given their relatively small spatial extent, medicanes tend to appear embedded in or to coexist with larger lows. Hence, the
10 implementation of a time-independent methodology avoiding the search of a medicane based on its location at previous time steps seems to be fundamental when facing situations of cyclones coexistence. The examples selected showcase how the algorithm presented throughout this paper is useful and robust for the tracking of medicanes. This methodology satisfies the requirements expected for a tracking method of this nature, namely: the capacity to track multiple simultaneous cyclones, the ability to track a medicane in the presence of an intense trough inside the domain, the potential to separate the medicane
15 from other similar structures handling the intermittent loss of structure, and the capability to isolate and follow the medicane center regardless of other cyclones that could be present in the domain. The complete TITAM package, including pre and post processing tools, is available as a free software extensively documented and prepared for its deployment. As a final remark, this algorithm sheds some light on the medicanes understanding, regarding the medicane structure, the warm-core nature and the existence of tilting.

# 1 Introduction

Cyclones can be broadly classified in terms of their thermal character as cold- and warm-core (Hart, 2003). Those developing in mid and high latitudes are cold-core, and obtain their energy from the baroclinic instability typical of these latitudes. Instead, warm-core cyclones develop in tropical and subtropical zones and, according to the latest theories (Zhang and Emanuel, 2016; Emanuel, 1986), are powered by enthalpy fluxes and maintained by self-induced heat transfer from the ocean (WISHE theory), where self-induced makes reference to winds associated to the cyclone. However, this conceptual framework, that considers two completely different types of storms, is a major simplification of real cyclones. Actual storms have a variable degree of similarity between these two idealised models, and indeed they evolve changing their thermal structure during their lifetime (Hart, 2003).

One particular case of storms are medicanes (from *Medi*terranean hurri*canes*), which do not perfectly fit any of these two idealised models. Medicanes are meteorological meso-scale systems formed in the Mediterranean basin, where baroclinicity provides the necessary atmospheric instability for the formation of cyclones. Under certain circumstances, the environment favours the tropical transition of the storm, then creating a spiral band of clouds around a well-defined cloud-free eye, while showing thermal symmetry and a warm core. The "tropical-like" term is introduced to account for the fact that, although they share similar mechanisms with tropical cyclones, they develop beyond the tropics (Homar et al., 2003; Gaertner et al., 2018).

According to the classical theory of tropical cyclone formation, a Sea Surface Temperature (SST) above 26º C (Palmén, 1948; Emanuel, 2003; Tous and Romero, 2013) is necessary for tropical cyclogenesis. In the absence of baroclinicity, a high SST is needed so that the lapse rate forces the atmosphere to be unstable enough for convection (Stull, 2017, Ch.16). However, the intrusion of a cold cut-off trough in upper levels, which causes cool air temperatures at high altitude, can trigger convection and lead to tropical cyclogenesis even when waters are not warm enough (McTaggart-Cowan et al., 2015). Hence, the fact that the presence of Mediterranean tropical-like cyclones is associated with cold-air intrusions explains why they can form even when the SST is below 26º C (Miglietta et al., 2011).

Midlatitude cyclones with tropical characteristics and actual tropical cyclones show similar but slightly different characteristics. Their main similarities are their appearance in satellite images, showing an eye in their structure, and their dynamical and thermodynamic features: a warm-core anomaly decreasing with altitude, weak vertical wind shear, strong rotation around the pressure minimum (high low-level vorticity) and convective cells organized in rain bands extending from the eyewall (Miglietta and Rotunno, 2019). The largest differences of medicanes with tropical storms pertain to the intensity and duration. Medicanes lifetime is restricted to a few days, due to the limited extent of the Mediterranean Sea, and they attain their tropical character only for a short period, while retaining extratropical features for most of their lifetime; the horizontal extent is generally confined to a few hundred km and the intensity rarely exceeds Category 1 of the Saffir-Simpson scale (Miglietta et al., 2011; Miglietta and Rotunno, 2019). Thus, while tropical cyclones can reach a radius of a thousand kilometers, 910 hPa of minimum central Sea Level Pressure (SLP) and 295 km $\cdot$ h$^{-1}$ per hour of maximum 1-minute sustained winds (Anthes et al., 2006; Shen et al., 2006), their Mediterranean counterparts show a smaller radius (up to 150 km) (Tous and Romero, 2013), a less intense central SLP minimum (980 hPa) and slower winds (gusts of about 180 km $\cdot$ h$^{-1}$) (Nastos et al., 2015; Miglietta et al., 2013).

In the literature, the detection and tracking methods for tropical cyclones is extensive. Some of them serve as a good base for the development of a medicane tracking algorithm, especially those applying a time independent methodology. Hodges (1994) presented a first work on an automated tracking method with general application to a wide range of geophysical fields. It is based on an identification of feature points by segmentation of structures and a further decomposition and analysis of the different structure points. The tracking part is based on former works (Salari and Sethi, 1990; Sethi and Jain, 1987) and consists in a constrained optimization of a cost function to determine the correspondence between the found feature points. Blender et al. (1997) succeeded introducing a time independent tracking method with few constrictions in order to allow a maximum applicability, including a further discussion on its validity for different spatial and temporal resolutions of the model data (Blender and Schubert, 2000). Vitart et al. (1997) also introduced an objective procedure for tracking-model-generated tropical storms similar to the one described by Murray and Simmonds (1991) using a time-independent approach. The same basic 2-steps methodology introduced in these works have been described in later works. Included among these are the one by Bosler et al. (2016), which addresses the issue of measuring distances at high latitudes, solving it by using geodesic distance instead of geometric distance between points. Also included in this set is one contribution by Wernli and Schwierz (2006), in which, besides a tracking algorithm, a new method for identifying cyclones and their extent is presented, being particularly useful for cyclonic climatological studies. Ullrich and Zarzycki (2017) argue that "uncertainties associated with objective tracking criteria should be addressed with an ensemble of detection thresholds and variables, whereas blind application of singular tracking formulations should be avoided", and provide a tool for tracking tropical and extratropical cyclones, along with easterly waves. Kleppek et al. (2008) employ "the standard method for midlatitudes" (Blender et al., 1997) and add the relative vorticity at 850 hPa to the center identification variables to address the difficulty of TC not being detected during genesis, decay or landfall stages. Other related works are the ones by Raible et al. (2008), which present a comparison of detection and tracking methods (Blender et al., 1997; Wernli and Schwierz, 2006; Murray and Simmonds, 1991) for tracking extratropical cyclones employing different reanalyses; by Zhao et al. (2009), adapting the earlier work by Vitart et al. (1997) and applying it for a climatology of global hurricane in a 50-km resolution GCM; or by Horn et al. (2014), who study the dependence of simulated tropical cyclone in climate model data on three tracking schemes (Walsh et al., 2007; Zhao et al., 2009; Camargo and Zebiak, 2002). It is also worth mentioning the contribution by Hanley and Caballero (2012), who succeed in the implementation of a novel method for recognizing 'multicentre cyclones', which is one of the main objectives of the present work, and even properly handling cyclone merger and splitting events; however, this method seems to rely solely on SLP, an important caveat when the objetive is the isolation of warm-core cyclonic systems. In general, although these methods are useful for tropical cyclones, even some of them being designed for a more general cyclone range, the particular case of medicanes shows important drawbacks, namely their coexistence with close extratropical lows, their temporal loss of the warm core nature due to a vertical tilting or their weak character when compared to genuine tropical cyclones.

Despite their similarities with tropical cyclones, there seems to be no agreement on the best algorithm to be used for the tracking of medicanes (Tous and Romero, 2013; Picornell et al., 2014). Concerning the medicanes tracking methods, some of them are designed to select a first track point, and calculate its movement direction from the different meteorological fields, along with some conditions that should be satisfied. This approach directly limits the applicability of the method, as it is

affected by a strong dependence on the initial tracking time (Hart, 2003). Thus, a time-independent tracking method seems

necessary for medicanes.

An additional problem is related to the detection of simultaneous storms at a time. While very uncommon, particularly when the considered domain is carefully chosen, the real coexistence in time of two medicanes inside a domain could happen, and the ability to capture more than one medicane may then be of utmost importance. Indeed, searching for two medicanes is technically the same as searching for two low pressure areas, and then the ability to handle multiple structures becomes

essential to avoid the risk of systematically tracking the one with the lowest pressure, instead of the one being medicane. The Hart parameters (Hart, 2003), which will be explained later, are derived variables used to characterize the thermal nature of the cyclone by means of the Hart conditions, used herein to find warm-core structures.

Thus, without overlooking the advantages of making progress in a precise medicane definition or the study of their genesis and maintenance dynamical and thermodynamical mechanisms, the main efforts of this work have been aimed at developing

a tracking algorithm allowing the coexistence of multiple storms of this nature. In this way, even in the absence of an optimal medicane definition, the flexibility provided by a parameter-oriented methodology favours the detection of this type of storm within a reasonable range of the parameters leading to that definition. As a previous step to introducing the designed algorithm, a brief review on the existing methods for tracking cyclones that are suitable for medicanes is carried out below and summarized in Table 1.

Picornell et al. (2001) introduce a widely used methodology for mesocyclones detection and tracking based on four steps: they first locate all the pressure relative minima as potential cyclones in each analysis, then filter them by imposing a minimum pressure gradient of 0.5 hPa/100 km at least along six of the eight directions; another filter based on the distance between two potential cyclones is applied too, taking the one with the largest circulation in case they are closer than four grid points; finally, they apply a methodology to calculate the track based on the hypothesis that the 700-hPa level is the steering level of the

movement of a cyclone (Gill, 1982), thereby considering the wind at that level to determine the direction in which the cyclone will preferably move. The methodology exposed in Alpert et al. (1990), based on a search for the track oriented within an ellipse whose major axis is defined by the 700-mb wind vector, is then extended with the definition of two additional elliptical areas in which the search of a storm center in the following time steps is performed. A disadvantage of this approach when applied to medicanes detection lies in the selection of a single point as medicane center before checking the warm/cold nature

of the cyclone. As we demonstrate below with an example, there may exist a little tilting in the medicane structure leading to a displacement between the points fulfilling the Hart conditions, detailed in Section 3.3.1, and the points showing the minimum surface pressure or cyclonic vorticity. If this were the case, then the track could suffer an artifactual loss of the tropical cyclonic nature when trying to impose the Hart conditions to the minimum pressure point of a tilted structure. This is also discussed in Hart (2003), along with the convenience of using either MSLP or vorticity for the identification of the cyclone center.

The methodology introduced by Hart (2003) has been widely applied in the years after its publication. It consists in making a time-dependent track by finding a first track point, and identifying the consecutive track points through a series of conditions based on center spatial and temporal displacement. Despite the difficulties that this method may face, its simplicity makes it very useful, and it has been used in this work as detailed below. In the same work, a phase space based on a set of parameters is

proposed to determine the thermal nature of a cyclone. These parameters are thoroughly revisited in this contribution and have

a great significance in the proposed method.

In a similar approach, Suzuki-Parker (2012) develop a tracking procedure dependent on the previous time step. The authors introduce previous filters by imposing thresholds in the 850 hPa wind speed, cyclonic relative vorticity and horizontal temperature anomalies.

Nevertheless, those algorithms based on the search of a new track point depending on the previous show important disad-

vantages for the purpose of tracking multiple cyclones at a time. Regardless of the criteria used to confine the search area for the next point, they are designed to find one single cyclone path, and show a strong dependence on the first chosen time step. In fact, this problem is clearly stated in Hart (2003), where he prevents the reader from this possible effect. The problem of tracking a cyclone by using its location in the previous time step is illustrated below through an example.

There exist more advanced tracking methods, such as the one suggested by Marchok (2002) based on Barnes interpolation

of seven different fields, namely the SLP, 700 and 850 hPa relative vorticities, 700 and 850 hPa geopotential heights and 2 secondary parameters (minimum in wind speed at 700 and 850 hPa). This method has been implemented as the operational NCEP cyclone tracking software.

Cavicchia and von Storch (2012) apply a tracking methodology founded on previous works (Zahn and von Storch, 2008b, a) and based on the identification of the pressure minima as potential centers and the subsequent clustering relying on the distance

between them. This method is very close to the one presented here in the concept of finding center candidates as independent entities, but shows a disadvantage: the pressure minimum, as shown below, is not always the best choice for the medicane center. A different field is here introduced with the purpose of preventing this pitfall. Besides, additional factors are considered to filter the center candidates, such as the Hart conditions or the symmetry of the geopotential height gradient.

Sinclair (1994) analyzes the limitation and benefits of using either SLP or vorticity for tracking. As detailed below, both

parameters are indeed used by the method we propose in this work to isolate the potential medicane centers.

Walsh et al. (2014) use both SLP and cyclonic vorticity to find medicane centers. Afterwards, temperature anomalies in the center are calculated to study the warm core nature of the cyclone. However, in the same way as in the previously mentioned methods, the selection of a single point could produce gaps in the tracks. This effect is acknowledged in their text and could be diminished by the multi-candidate selection and clustering method proposed here.

Here a new methodology for tracking medicanes is presented. It overcomes the drawbacks of previous methods. This new methodology does not need an initial state of the medicane, is able to identify various simultaneous structures and prevents the aforementioned loss of structure. Besides, its parallel performance (see Appendix C for details) enables its application to long term simulations.

## 2 Preprocessing: building the input data

The total tracking procedure consists of a first step for preparing the input data, a second step with the execution of the algorithm and a final postprocessing of the output data provided by the algorithm.

The input data of the algorithm consists of files containing temporal series of a number of meteorological fields. The mandatory 2D and 3D fields are SLP, 10-m wind horizontal components (U10, V10) and geopotential height (Z) for, at least, the 900, 800, 700, 600, 500, 400 and 300 hPa levels.

The input provided by the user must be compliant with the specifications given in the Appendix B, regarding the input format, the internal name of the variables and dimensions, the physical units and the matrices order. Note that the algorithm package includes a pre-processor for Weather Research and Forecasting (WRF) model output called 'pinterpy' (more details in Appendix B).

## 3 Medicane tracking algorithm

The TITAM algorithm is rather complex and consists of several steps, so the main components are briefly outlined here, while the details of each part are thoughtfully described in the following subsections (Figure 1). Overall, the algorithm can be divided in two main blocks: the detection of the cyclone (medicane) centers in each time step (red box in Figure 1), and the creation of a track by joining the centers through the time domain (D).

The detection block consists of three main steps. In the first part, (A) the algorithm makes a first selection of the potential candidates to medicane centers. Once the candidates are selected, (B) they are grouped using an ad-hoc clustering method. Each group eventually leads to a potential cyclone. Finally, (C) the algorithm searches for a center of the cyclone verifying the thermal conditions for being a medicane, i.e. the Hart conditions, explained below. The search of centers is carried out for each time step separately and regardless of their location in previous step. This allows us to benefit from a key feature of the algorithm: time independence. It enables a straightforward parallelization in the code implementation (see Appendix C for details).

In the second block (D), the points resulting from the procedure above, which are not yet connected in space or time, get linked following a set of rules. The details are given in Section 3.4.

### 3.1 Searching for center candidates (A)

#### 3.1.1 Filtering by cyclonic potential, SLP and vorticity

The first step is to define a diagnosed field acting as an indicator of areas with high vorticity and exposing a minimum in the pressure field, i.e. those prone to cyclonic activity. The selected variables are 10-meters relative vorticity and SLP laplacian. Using the product of these two fields emerges as a good strategy for finding the candidate points. This magnitude brings out all the points being SLP minima with high cyclonic character. This diagnosed field, hereafter referred as cyclonic potential $\mathcal{C}$, is thus defined as:

$$\mathcal{C} = \nabla^2(\text{SLP}) \cdot (\boldsymbol{\nabla} \times \boldsymbol{v_{10}})_z \tag{1}$$

where the dot represents a Hadamard product, and the $z$ subscript means that only the z-component of the surface wind curl (i.e., the surface vorticity) is considered. Given this definition, a high positive value of $\mathcal{C}$ at a given point exposes the cyclonic

nature of the flow around it. The definition of $\mathcal{C}$ is motivated by the relationship between the geostrophic relative vorticity and the laplacian of the pressure field obtained within the context of the quasi-geostrophic theory; this is

$$\xi_g = \frac{1}{\rho_0 f}\nabla_h^2 p \qquad (2)$$

where $\xi_g$ is the geostrophic relative vorticity; $\rho_0$ and $f$ are constants, and $\nabla_h$ is the horizontal gradient operator at fixed height (Holton and Hakim, 2012). Hence, the product represented by $\mathcal{C}$ would be redundant if the 10-meters wind field was well-represented by the geostrophic wind approximation at surface level. Nevertheless, for a medicane, large surface effects are present and the surface wind is thus not well represented by the geostrophic approximation. Indeed, from this point of view this product is expected to report a greater benefit with respect to using the SLP alone in those cases where SLP perturbations occur due to orographic factors.

Once $\mathcal{C}$ is calculated, this field is successively 1-2-1 smoothed $N$ times (see parameter *SmoothingPasses* in Appendix A). This filter is necessary because of the noisy character of the SLP Laplacian in high resolution data. The next step is to filter out all the grid points with a SLP value above a certain threshold (see parameter *SLPThreshold* in Appendix A) and those with a $\mathcal{C}$ value above the threshold marked by a given percentile (99.9 by default, see parameter *ProductQuantileLowerLimit* in Appendix A) are retained. On the other hand, a review of the vorticity values exhibited in the different medicanes simulations suggest that a lower threshold of 1 rad $\cdot$ h$^{-1}$ is enough to filter out the situations where no medicanes are present (see parameter *VorticityThreshold* in Appendix A). Therefore, points with lower cyclonic potential are removed following the above criteria. Note that, provided the definition of vorticity, it is dependent on the horizontal grid spacing and, henceforth, the provided default value for the vorticity threshold may not be suitable for cases with different horizontal grid spacings.

### 3.1.2 Symmetry and radius

The next step consists in applying a filter to remove candidates to cyclone center based on the symmetric structure and radius of the medicane. Any point not satisfying both conditions is discarded as center candidate. The horizontal domain of a cyclone is defined as the area of positive vorticity around the cyclone centre, bounded by the zero-vorticity line (Picornell et al., 2001; Radinovic, 1997). This domain, which should be quasi-symmetric in the case of a medicane, is used to define the medicane effective radius (MER). The zero-vorticity radius is defined as the distance from the candidate point to the points where vorticity changes its sign, from positive to negative (see parameter *CalculateZeroVortRadiusThreshold* in Appendix A). In our case it is calculated for eight angular directions (every $\pi/4$ radians). The MER is then estimated as the mean of the eight zero-vorticity radiuses. This calculation is conditioned by the number of points considered for performing the sign change search over each direction, which is equivalent to the maximum distance tested (see parameter *CalculateZeroVortRadiusDistance* in Appendix A).

Conditionally (see parameter *IfCheckZeroVortSymm* in Appendix A), we can check the symmetry of the zero-vorticity line. Firstly, we impose the requirement that the zero-vorticity radius must exist for a minimum number of the eight directions tested (see parameter *ZeroVortRadiusMinSymmDirs* in Appendix A). Next we define the asymmetry coefficient $A_c$ as the maximum difference of the eight calculated radiuses. The candidate point is rejected as such if $A_c > A_p$, where $A_p$ is an

algorithm parameter (see parameter *ZeroVortRadiusMaxAllowedAsymm* in Appendix A). Finally, to keep the candidate point, we impose the calculated MER to be in a range of possible radiuses, maximum $\text{MER}_H$ and minimum $\text{MER}_L$ (see parameters *ZeroVortRadiusUpperLimit* and *ZeroVortRadiusLowerLimit* in Appendix A). These parameters must be set by the user based on the typical observed values for medicane MERs. The points discarded by this filter are mainly orographic artifacts which tend to appear due to orography-induced vorticity. Note that this condition of symmetry of the zero-vorticity radius is similar to that of SLP gradient in multiple directions used by other authors (e.g. Picornell et al., 2001; González Alemán, 2019; Cavicchia and von Storch, 2012). The main difference lies in the fact that they impose a lower limit for the SLP gradient in the different directions, but do not check the difference in magnitude across gradients.

A consistent calculation of this zero-vorticity radius is of great importance, as it will serve as the radius to calculate the Hart parameters to the points held as center candidates after the filters. Defining a variable radius which depends on the situation rather than a constant unique value is a flexible solution that overcomes the problem of dealing with very different structures in the same domain (Cioni et al., 2016; Picornell et al., 2014; Chaboureau et al., 2012; Miglietta et al., 2011).

## 3.2 Grouping potential centers (B)

As previously mentioned, the advantage of allowing multiple center candidates is the possibility of finding a medicane center not being neither the absolute SLP minimum nor the point with maximum value of $\mathcal{C}$, as those could not fulfill the thermal structure of warm-core cyclones. On the other hand, the algorithm should ideally have the ability of finding multiple concurrent cyclones. To achieve these requirements, we separate the center candidates into different clusters. Note that the number of points passing the previous filters must be above the number of points marked by the parameter *MinPointsNumberInCluster*.

The cluster classification is built upon a distance $d_c$ that marks the minimum separation distance between two cluster representative points (see parameter *SLPminsClustersMinIBdistance* in Appendix A). This parameter should be set having into account the common range within which a medicane radius usually lies. The clustering method is a reduced k-means clustering without iterative calculation, in which the number of groups (see parameter *MaxNumberOfDifferentClusters* in Appendix A) is computed as the number of center candidates separated by more than the distance $d_c$ from the other candidates. The cluster centers are selected by $\mathcal{C}$ value: the point with the highest $\mathcal{C}$ is selected as center, the second one is selected as center if their distance is higher than *SLPminsClustersMinIBdistance*, and so on. Imposing an upper limit for the number of clusters prevents the inclusion of clusters not being real medicane candidates in large domains, especially if the values selected for the previous filters were not tight enough.

The final task of the grouping method is to filter out all the points belonging to clusters formed by less than a minimum number of points (see parameter *MinPointsNumberInCluster* in Appendix A). These clusters are considered to be too small to constitute a medicane structure and, hence, their points are discarded as center candidates.

### 3.3 Identification of warm core structures (C)

The final list of center candidates is composed by those points which pass all the filters and conditions, showing a high cyclonic character and a high symmetry in the zero-vorticity line enclosing the medicane domain, as well as pertaining to a cluster made up of enough candidates to be considered as a medicane structure.

#### 3.3.1 Hart conditions

The thermal nature of a cyclone is customarily studied through the so-called Hart parameters (Hart, 2003). Based on these parameters, the Hart conditions are described regarding the existence of a thermal symmetry around the center, and the warm core character of the cyclone nucleus. These two features define the nature of a tropical cyclone. The former is evaluated by means of a symmetry parameter $B$, defined as:

$$B = h \left( \overline{Z_{600\,\text{hPa}} - Z_{900\,\text{hPa}}}\big|_R - \overline{Z_{600\,\text{hPa}} - Z_{900\,\text{hPa}}}\big|_L \right) \tag{3}$$

where h = +1 for the northern hemisphere, and -1 for the southern one. $B$, measured in metres, relates to the thermal symmetry around the core of the cyclone, with warm-core cyclones being highly symmetric. The horizontal bar denotes a spatial average over all the points on a specific side of a circle with center in the cyclone center, and radius $R_B$. The MER value is used for $R_B$ in this algorithm.

Hart (2003) states that a threshold of 10 meters marks the existence of thermal symmetry. However, in case of non-symmetric systems, there is a strong dependence on the section used to divide the circle. Hence, even though the original definition of $B$ is based on a single left-right section over the cyclone motion, the proposed method in this paper is more general and flexible allowing the calculation of a mean $B$ parameter over four different directions to remove the possibility of the cyclone motion direction being a privileged one. This is necessary to cope with the structure of medicanes, which is not as clearly symmetric as in the case of tropical cyclones.

Some studies (see, e.g., Picornell et al., 2014) have discussed the radius over which this spatial average should be performed, as well as the pressure levels that define the layer thicknesses. The original radius value suggested by Hart (2003) is 500 km, but a lower value must be set for medicanes having into account their smaller size respect to tropical cyclones.

The warm core nature of a cyclone is directly related, by the thermal wind relation, with the shear of the layer thickness. Therefore, Hart (2003) defines a modified thermal wind as:

$$-|V_{T_L}| = \frac{\partial \left( \frac{\Delta Z}{d} \right)}{\partial \ln p} \bigg|_{900\,\text{hPa}}^{600\,\text{hPa}} \tag{4}$$

$$-|V_{T_U}| = \frac{\partial \left( \frac{\Delta Z}{d} \right)}{\partial \ln p} \bigg|_{600\,\text{hPa}}^{300\,\text{hPa}} \tag{5}$$

where the $L$ and $U$ subscripts denote the lower and upper tropospheric layers, respectively, and $d$ accounts for the different distances between the geopotential extrema inside a pressure level for the different pressure levels. There is an open question

about the appropriate values of the pressure levels limiting the upper troposphere and lower tropospheric layers when studying medicanes. Here the same levels as in Hart (2003) are used. 900 hPa is selected as the lower troposphere limit, and 300 hPa as the level close to the tropopause. 600 hPa level divides the 900-300 hPa layer in two atmospheric layers with equal mass. As defined here, the thermal wind is in fact a dimensionless scaled thermal wind.

285     As described by Hart (2003), the existence of a warm core cyclone directly results in both $-|V_{T_L}|$ and $-|V_{T_U}|$ being positive, the former being usually greater in magnitude than the latter one. These three conditions are thus imposed as part of the algorithm at each center candidate to ensure the warm-core of the environment around these points before selecting them as actual medicane centers.

### 3.3.2   The Hart-checking for the identification of a warm-core structure

290   The Hart parameters provide a phase space for an objective classification of the cyclones according to their thermal structure into tropical and extratropical cyclones. It is a common practice (see, e.g., Miglietta et al., 2011; Cioni et al., 2016) to analyse the phase space of the cyclone after having identified its track. However, it could be the case that we defined a center for the system, used it to define the tracking of the storms, and it turned out that this grid point does not fulfill the specific requirement of being the center of a warm-core storm. To prevent this behaviour, not uncommon in storms where the thermal character is 295 not so strongly defined as in the case of tropical cyclones (we illustrate this with an example in Section 4.1), we reverse the order: checking the Hart conditions before selecting a point as medicane center.

    If the parameter *IfCheckHartParamsConditions* is set to false, then the point with the minimum SLP value of each cluster will be selected as the center. Otherwise, the Hart conditions are checked over the cluster points to select the center. For the Hart-checking of the points, multiple parameters can be tuned (see Appendix A) regarding the Hart conditions to 300 check (*HartConditionsTocheck*), the pressure levels related to the Hart parameters calculation (*Blowerpressurelevel*, *Bupperpressurelevel*, *LTWlowerpressurelevel*, *LTWupperpressurelevel*, *UTWlowerpressurelevel* and *UTWupperpressurelevel*), or their thresholds (*Bthreshold*). In particular, the $B$ parameter calculation is slightly different from that proposed by Hart (2003), and is extended to check the layer thickness symmetry in multiple directions, relying on the parameters *Bmultiplemeasure* and *Bdirections* (see Appendix A).

305   Thus, for each cluster, its center candidates are sorted by the SLP value. Hart conditions are calculated for each point until one of them fulfills the Hart conditions. Either this happens, or all the points inside a cluster are Hart-checked without any point meeting the Hart conditions, the same procedure is applied to the next cluster until no clusters are left.

### 3.4   Postprocessing: Building the track (D)

Once the medicane centers have been identified for each time step according to the criteria explained in the former section, 310 the next algorithm component connects such points to generate the cyclone track. The reconstruction of the cyclone path from disjointed points is based on the connection of two medicane centers found at different time steps. Define two parameters, namely the maximum spatial separation ($D_{max}$, in kilometers) and the maximum temporal separation ($DT_{max}$, in time steps) between two points to be connected. Let $M_t^c$ be the location of the medicane center at time $t$ and $M_{t'}^c$ the location at time $t'$: if

$t' - t \leq DT_{max}$ and $M_{t'}^c - M_t^c \leq D_{max}$, then $M_t^c$ and $M_{t'}^c$ are connected. In the case of $DT_{max}$ being higher than one time step, two points $M_t^c$ and $M_{t'=t+DT_{max}}^c$ are connected if the following is true: $\nexists$ i, i $\in \mathbb{N}$, i $< DT_{max}$: $M_{t+i}^c - M_t^c \leq D_{max}$. This prevents a point from being connected at the same time with multiple previous centers if $DT_{max}$ is chosen to be greater than one time step.

This connected track can be overlaid to a map with the correct projection corresponding to that of the input data by using the plotting tool provided in this package, as described in the Appendix D. Besides, multiple measures of the medicane size and intensity along its path can be obtained by means of another tool (*getmedicanestrackdata*) contained in this package (see Appendix D for further information).

## 4 Testing the algorithm

In this section, four examples of the application of the algorithm are put forth to showcase its properties and capabilities. First, we will show how the algorithm works step-by-step for a canonical case: the Rolf medicane. The second example verifies the suitability of the algorithm to differentiate between tropical and extra-tropical cyclones. The third example will show the advantages of not using the minimum pressure as a monitoring method as well as the independence of the initial tracking time. The last example shows the ability of the algorithm to distinguish and track two simultaneous medicanes.

Most of the shown examples consist of experiments performed with the Weather Research and Forecasting (WRF) model driven by ERA interim reanalysis data. Details about the simulations carried out can be found in Appendix E.

### 4.1 The case of the Rolf medicane

This case study represents a canonical medicane event, the Rolf medicane. It is the longest-lasting and probably the most intense medicane ever recorded in terms of wind speed (Kerkmann and Bachmeier, 2011; Dafis et al., 2018), and will therefore serve as a good testbed (Ricchi et al., 2017) for presenting a step-by-step review of the algorithm. The data analyzed comes from a numerical simulation at 9 km of grid spacing (see Appendix E for details). The simulated period extends from 2011-11-05 to 2011-11-10 with hourly temporal resolution.

Figure 2 (bottom) shows an example of the cyclonic potential field $\mathcal{C}$, used in the first place to select the candidate points, for a given time (2011-11-07T23:00). The SLP laplacian (top panel) is noisy and mostly driven by orography, while wind curl (middle panel) is highly prone to suffer orographic effects. The cyclonic potential $\mathcal{C}$ (bottom panel) significantly reduces noise, and its smoothing results in a clearer picture of the potential medicane locations.

Once the cyclonic potential is calculated, the center candidates are selected by imposing the conditions described in Sections 3.1.1 and 3.1.2 (the default values for all the parameters are used, see Appendix A). The points selected as center candidates (56) are represented in the bottom plot of Figure 2 with black crosses. Note that, given the intensity and well-defined symmetric shape of the medicane, all the points selected by the percentile are inside the medicane domain, and none are filtered out by the conditions. In this case, given its small domain extent, all the points are grouped within a single cluster. Finally, the centers inside the cluster are reordered by SLP value, and the Hart parameters are calculated until a center is found.

As commented above, the medicane center selected does not necessarily coincide with the SLP minimum. This is particularly true when the SLP minimum does not satisfy the Hart conditions or any of the conditions imposed before. This is clearly illustrated in Figure 3, where the bottom panel represents the fulfillment of the Hart conditions by the SLP minimum (not the absolute one, but that inside the zero-vorticity domain, which is selected as medicane center if it fulfills the Hart conditions) and the center selected by the algorithm. A filled circle indicates that the point meets the Hart conditions, and its colour is related with the SLP value. The other symbols indicate the Hart condition infringed by the SLP minimum point when it does not coincide with the medicane center found by the algorithm. Top panel represents the Hart phase space plots for both sets of data. As expected, the algorithm classifies much more time steps as medicane than those obtained by using only the SLP minimum. Furthermore, from the top panels we can conclude that, most of the times, it is the symmetry condition for the geopotential height thickness the one preventing the SLP minimum point from fulfilling the Hart conditions and, hence, from being selected as medicane center.

In addition, Figure 4 shows a complete trajectory of the Rolf medicane as tracked by the algorithm presented here, along with the SLP relative minimum found at each time step in the proximity of the medicane center. This track is the result of passing the complete algorithm to the simulation with the default values of the parameters, as presented in Appendix A. When there is no coincidence between the SLP minimum and the found center for the medicane, marked in blue, it means that the SLP minimum does not fulfill the Hart conditions, and is coloured in red. Conversely, a green dot marks the SLP minimum for the time steps in which it fulfills the Hart conditions and is selected as the medicane center.

Therefore, we obtain that the center of the medicane does not coincide with the SLP minimum for the conditions imposed (see table in Appendix A for further detail) for a large portion of the time steps. Hence, tracking the SLP minimum and checking Hart conditions after the tracking method would result in a loss of the medicane character for a majority of time steps. In this sense, the obtained tracking is almost point-by-point connected (a medicane is found in almost every time step) and thus more robust. This behavior can be attributed to the tilting of the medicane core. In Figure 5, we compare the medicane structure for two different time steps. The structure is represented, in the left side, by the cross section of the equivalent potential temperature $\theta_e$ (colors), the SLP (dashed grey line) and the geopotential height thickness ($Z_{600} - Z_{900}$), both scaled to the zero-one interval (unity-based normalization). The right side of the figure corresponds to a spatial latitude/longitude projection of the SLP (colours) and the geopotential height 600-900 hPa layer thickness (dashed contours). In the first case, corresponding to 2011-11-07T16:00 (top panel of Figure 5), the relative SLP minimum among the points of the medicane activity area is within the highest ($Z_{600} - Z_{900}$) layer thickness isoline (right), being the medicane center coincident with the point showing the lowest SLP value. In addtion, the cross section reveals a perfect correspondence between SLP minimum and layer thickness maximum, as well as a great symmetry of $\theta_e$ around the vertical axis traced through the medicane center. This is related with a non-tilted medicane core.

Conversely, in the second case, corresponding to 2011-11-08T23:00 (bottom panel of Figure 5), the medicane center detected by the algorithm is not coincident with the SLP minimum. The SLP minimum is almost out of the highest thickness contour and is 30 km away from the medicane center (about a 30% of the medicane radius). The value of the Hart $B$ parameter for the medicane center (dotted black vertical line) is 9 m, while for the SLP minimum at the same latitude of the actual medicane

center is 20 m. Note that the medicane center is coincident with the maximum value of thickness. For this time step, the $\theta_e$ vertical pattern does not show symmetry around the axis, but a tilting of the medicane core.

Therefore, the high capacity of our algorithm to detect medicanes is mainly based on the ability to recognize situations where the medicane presents a slightly tilted structure. This tilting is not present in tropical cyclones and is what leads medicanes to easily lose their structure, thus encumbering the task of medicanes tracking.

## 4.2 A deeper low in the domain

In the way the algorithm was conceived and developed, it should be able to isolate medicane structures even in the presence of a deeper low in the domain. In order to verify this ability, a simulation of the Rolf medicane is run with a domain extending to high latitudes, where the development of deep lows is very common. To reduce the computational cost of the simulation, and to test the algorithm with fields of coarser grid spacings, this simulation is run at 27 km (see details in Appendix E).

Figure 6 (top) shows the SLP field for the whole domain at 2011-11-07T12:00. The synoptic situation is characterized by a deep extratropical cyclone located at the North Atlantic, being the pressure center lower than 980 hPa. Simultaneously in the Western Mediterranean Sea, a potential medicane (Rolf) appears with a pressure center around 1000 hPa. Figure 6 (bottom) shows the cyclonic potential $\mathcal{C}$ for the same time step. In this first algorithm step we see how both cyclones are isolated, specially highlighting the medicane structure. High vorticity values are also present associated with the cold front in the Atlantic low. In the second step (Figure 7 left), the quantile filter (black crosses) and the vorticity threshold filter (red crosses) are applied. In the next step (Figure 7 right), the points with the required zero-vorticity radius symmetry are selected (blue crosses). Therefore, at this point we have two clusters with several medicane center candidates, whose representative points (highest $\mathcal{C}$ valued points) are represented as large red plus symbols (one for the Atlantic low and one for the Mediterranean low) in Figure 7 (right).

Finally, the algorithm results for this time step show how there is no point fulfilling the Hart Conditions in the Atlantic low, while it correctly finds a medicane center in the Mediterranean low (green plus in right plot of Figure 7). Therefore, the algorithm successfully achieves the desired isolation of the medicane despite of the presence of a deeper low within the domain. The final track obtained is presented in Figure 8 (blue line). The domain is cropped to the Western Mediterranean area given that no medicane center is found by the algorithm for the Atlantic low. In addition, the ability of the algorithm to assimilate and handle several sources of data is also illustrated. The track of the medicane from the ERA5 reanalysis as calculated by the algorithm over a similar spatial domain is also presented (Figure 8, dark red line).

## 4.3 Medicane independence from the low pressure center

As previously stated, an important drawback of algorithms based on the search of new track points depending on previous ones lies in its strong dependence on the selection of the first time step, regardless of the criteria used to confine the search area for the subsequent point.

For illustrating this problematic situation, we select a 9 km WRF simulation of the Celeno medicane (see Appendix E for details). The simulation reproduces the generation of the medicane. Although the obtained track does not fit the one reported by former studies (Pytharoulis et al., 1999; Lagouvardos et al., 1999), this simulation still seems valid for testing the algorithm.

The meteorological situation is characterized by an eastward-moving extratropical cyclone (see Figure 9) detected on 1995-01-13T08:00 and traveling until 1995-01-14T09:00 as far as the north of the Libyan coast. During the 1995-01-14 morning appears a strong cyclogenetic character within an area around the Ionian Sea (see Figure 9), emerging a medicane at 1995-01-14T14:00 that travels first to the west and turns to the south-east. Finally the medicane reverses into an extra-tropical cyclone travelling throughout the Eastern Mediterranean Sea.

Therefore the model reproduces a situation where two main lows coexist in the domain for a few hours (Figure 9). Using a time dependent algorithm, if it started tracking in the time step shown in panel A, the initial point would correspond to the the minimum SLP (labelled as 'CY'). Tracking this point would lead to follow one low that will not satisfy the warm-core conditions, being the medicane (with the 'ME' label) located 400 km away from the actual cyclone (panel D). Then, while the former is more intense in terms of SLP minimum, it is the latter one that fulfills the conditions to be a medicane. The algorithm does not follow the synoptic low ('CY') since it does not satisfy other conditions such as the symmetry (Figure 10).

This example shows how a time independent method provides the algorithm with the capability of tracking several lows, which in certain circumstances is necessary to permit a correct detection of the medicane.

### 4.4 Coexistence of two simultaneous medicanes

One remarkable feature of this algorithm is its ability of capturing several simultaneous warm core structures. In this section we present the application of the algorithm to a 9 km WRF simulation of the Leucosia medicane event. The simulation period was 1982-01-19 to 1982-01-28. More details about the experiment can be found at Appendix E. Although there is no evidence that this event showed two simultaneous medicanes (Ernst and Matson, 1983; Reed et al., 2001), the simulation reproduces them. Therefore it serves as a particularly interesting trial for the algorithm, given that the algorithm implementation allows the parameters tuning to search other types of cyclones more likely to coexist in the same domain.

The simulation reproduces the formation of two coexisting medicanes during a period of 24 hours. Figure 11 presents the tracks detected by the algorithm for the whole simulated period and the SLP field for a time where both warm core structures coexist. The track located at the north of Libya corresponds to the documented tropical-like cyclone event Leucosia, which maintained its medicane characteristics from early 25 to mid 26 of January. Another tropical-like cyclone coexisted with Leucosia for 24 hours since 1982-01-25T04:00 and faded after reaching the Apulia region of Italian Peninsula.

While this may seem a situation not prone to happen, the interesting point here is that the algorithm is prepared to avoid the Hart conditions and track regular cyclones. Since unlike two medicanes, the coexistence of two cyclones in general is a very common event, we remark here the ability of the algorithm to track simultaneous storms.

### 5 Conclusions

In this work, a new algorithm specifically suited for medicanes tracking has been presented. The algorithm is robust and capable to detect and track them even in adverse conditions, such as the existence of larger or more intense systems within the domain, the coexistence of multiple tropical-like systems or the existence of complex orographic effects. This algorithm

implements a time independent methodology whose search methodology does not rely on previous time steps, and hence the time independence. Although it is especially suited for medicanes, it also provides the possibility of an easy modification of the cyclone definition parameters to make it useful for the detection of different cyclone types.

The algorithm mainly bases on a cyclonic potential field $\mathcal{C}$, and the method applies successive filters over all grid points on each timestamp, leading to a final list of center candidates. After grouping them to allow the existence of multiple cyclones in the same domain, the Hart conditions are used to select a single center within each cluster of candidates, i.e. for each medicane structure. Eventually, the found centers are connected over time and space, and a complete medicane track is obtained as the main product of the algorithm. The computational efficiency and time-saving performance have been key factors taken into account for the development of this algorithm. Consequently, it should be suitable for further medicanes climatological studies.

The selected examples showcase how the algorithm presented throughout this paper is useful and robust for the tracking of medicanes. The tracking algorithm allows to detect these storms and even in the weakest phases of the weakest events, differentiating this type of storms from mid-latitude cyclones. This methodology satisfies the requirements expected for a tracking method of this nature, namely: the capacity to track multiple simultaneous cyclones, the ability to track a medicane in the presence of an intense trough inside the domain, the potential to separate the medicane from other similar structures handling the intermittent loss of structure, and the capability to isolate and follow the medicane center regardless of other cyclones that could be present in the domain.

The use of TITAM for the automatized detection of other types of cyclones, or even for the detection of medicanes on early or late stages, can be easily achieved by modifying the Hart conditions module within the algorithm namelist. When ignoring the Hart conditions, the selected center represents the point with the lowest SLP value among the points with the highest $\mathcal{C}$ value fulfilling the zero-vorticity radius symmetry condition. This is virtually equivalent to track the SLP minimum along its motion, as long as it fulfills the zero-vorticity radius symmetry condition. Despite its complexity due to the existence of multiple parameters, the namelist-oriented implementation provides it with the flexibility needed to apply it to the tracking of other kinds of cyclones. Thus, it is an extensible tool that can be used for the automated identification of medicanes and other types of cyclones (tropical and extratropical) in large datasets such as in regional climate change experiments. The complete TITAM package is available as a free software extensively documented and prepared for its deployment (see *Code availability* below).

As a final remark, this algorithm sheds some light on the medicanes understanding, regarding the medicane structure, the warm-core nature and the existence of tilting.

*Code availability.* The code developed to build up the TITAM algorithm is fully available as an open-access resource (Pravia-Sarabia et al., 2020) at Zenodo database. Bash scripting has been used to weave R functions into an usable user-friendly package. Final product is a set of Bash scripts conceived for a namelist-oriented usage. The *pinterpy* interpolation tool is based on the Python library wrf-python (Ladwig, W. (2017). wrf-python (Version 1.3.0.) [Software]. Boulder, Colorado: UCAR/NCAR. https://doi.org/10.5065/D6W094P1). Figures have been prepared with R software.

*Data availability.* All the WRF simulations presented in this paper as object of the algorithm testing procedure have been carried out in the MAR group of the University of Murcia. The simulations output data, as well as the files of ERA5 reanalysis data used to generate figures presented throughout this paper, are available as an open-access resource (Pravia-Sarabia, 2020) at Zenodo database. ERA-Interim reanalysis data used as WRF model input can be downloaded from the Copernicus Climate Change Service Climate Data Store (CDS). .

*Author contributions.* EPS carried out the WRF simulations, wrote the algorithm code and performed the calculations of this paper. JPM contributed to the design of the simulations and their analysis. He also provided ideas for new approaches in the analysis of the simulations that have been integrated in the final manuscript. JJGN, PJG and JPM provided substantial expertise for a deep understanding on the medicanes concept, which led to a successful conception of the algorithm. The paper has been written by EPS, JPM and JJGN, and all authors have contributed reviewing the text.

*Competing interests.* The authors declare that they have no competing interests.

*Acknowledgements.* This study was supported by the Spanish Ministry of the Economy and Competitiveness/Agencia Estatal de Investigación and the European Regional Development Fund (ERDF/FEDER) through project ACEX-CGL2017-87921-R. Juan José Gómez-Navarro acknowledges the CARM for the funding provided both through the Seneca Foundation (project UMULINK 20640/JLI/18), as well as the "Juan de la Cierva-Incorporación" programme (IJCI-2015-26914).

| Method | SLP | Vorticity | Axisymmetry check | Time independent | Hart conditions | Spatial distance | Temporal distance |
|---|---|---|---|---|---|---|---|
| Alpert et al. (1990) | Yes | No | No | No | No | Yes | Yes |
| Picornell et al. (2001) | Yes | No | Yes | No | No | Yes | Yes |
| Hart (2003) | Yes | No | No | No | No | Yes | Yes |
| Suzuki-Parker (2012) | Yes | Yes | No | No | Yes | Yes | Yes |
| Marchok (2002) | Yes | Yes | No | Yes | No | Yes | Yes |
| Cavicchia and von Storch (2012) | Yes | No | Yes | Yes | No | Yes | Yes |
| Zahn and von Storch (2008b) | Yes | No | No | Yes | No | Yes | Yes |
| Sinclair (1994) | No | Yes | No | No | No | Yes | Yes |
| Walsh et al. (2014) | Yes | Yes | No | No | Yes | Yes | Yes |
| TITAM | Yes | Yes | Optional | Yes | Optional | Yes | Yes |

**Table 1.** Summary of some cyclone tracking methods usually applied to medicanes.

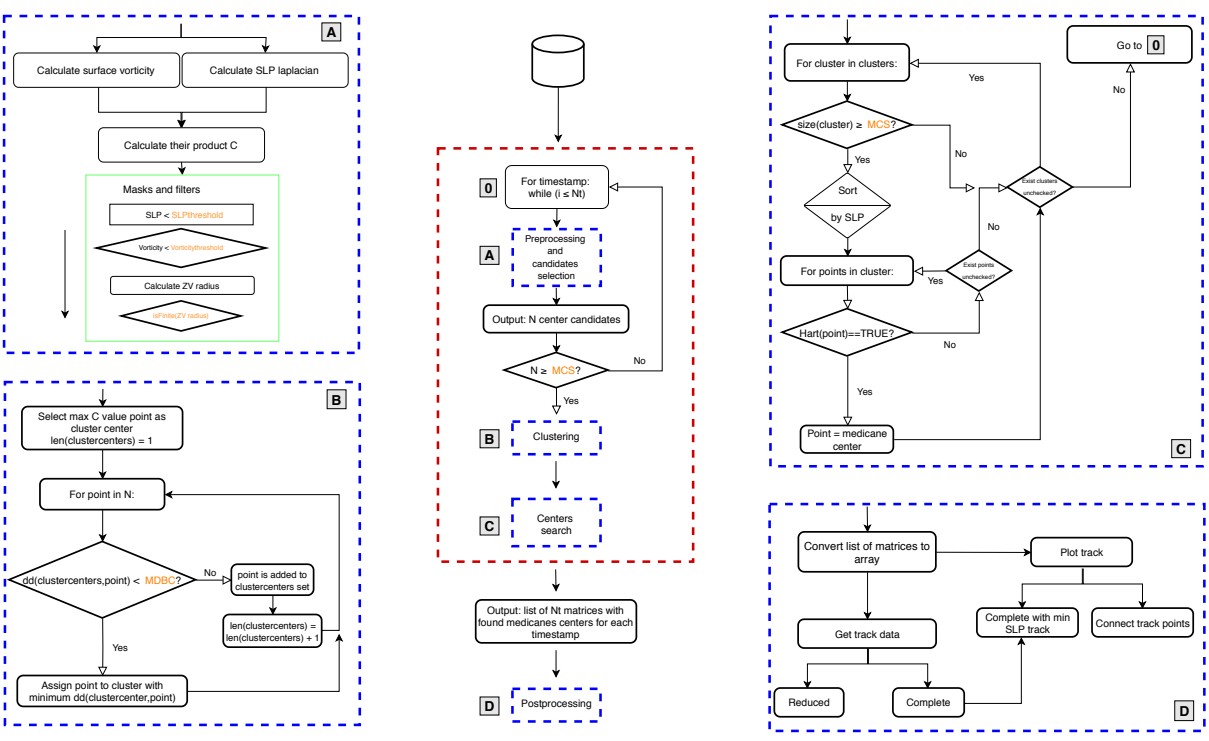

**Figure 1.** Flow chart describing the algorithmic implementation for the proposed medicane detection methodology TITAM (for the medicanes detection part). MCS and MDBC correspond to the *MinPointsNumberInCluster* and *SLPminsClustersMinIBdistance* algorithm parameters, respectively.

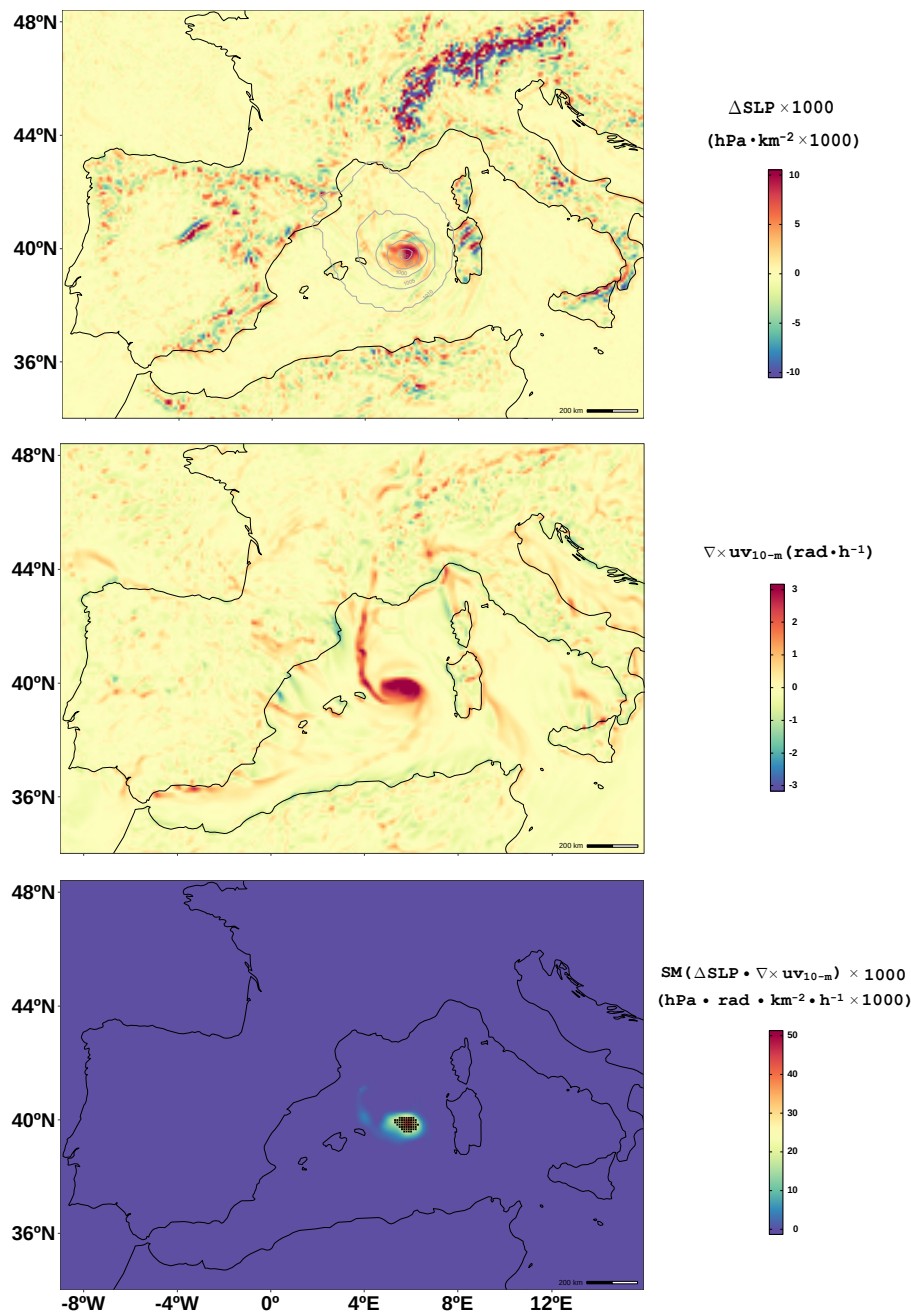

**Figure 2.** Three fields derived from the Rolf simulation with 9 km of grid spacing (see Appendix E). The SLP laplacian is shown in colours along with SLP contours coloured in grey (top panel); the 10-m wind curl and cyclonic potential $\mathcal{C}$ are presented in the middle and bottom panels, respectively. Black crosses in bottom plot represent points selected as center candidates before checking Hart conditions.

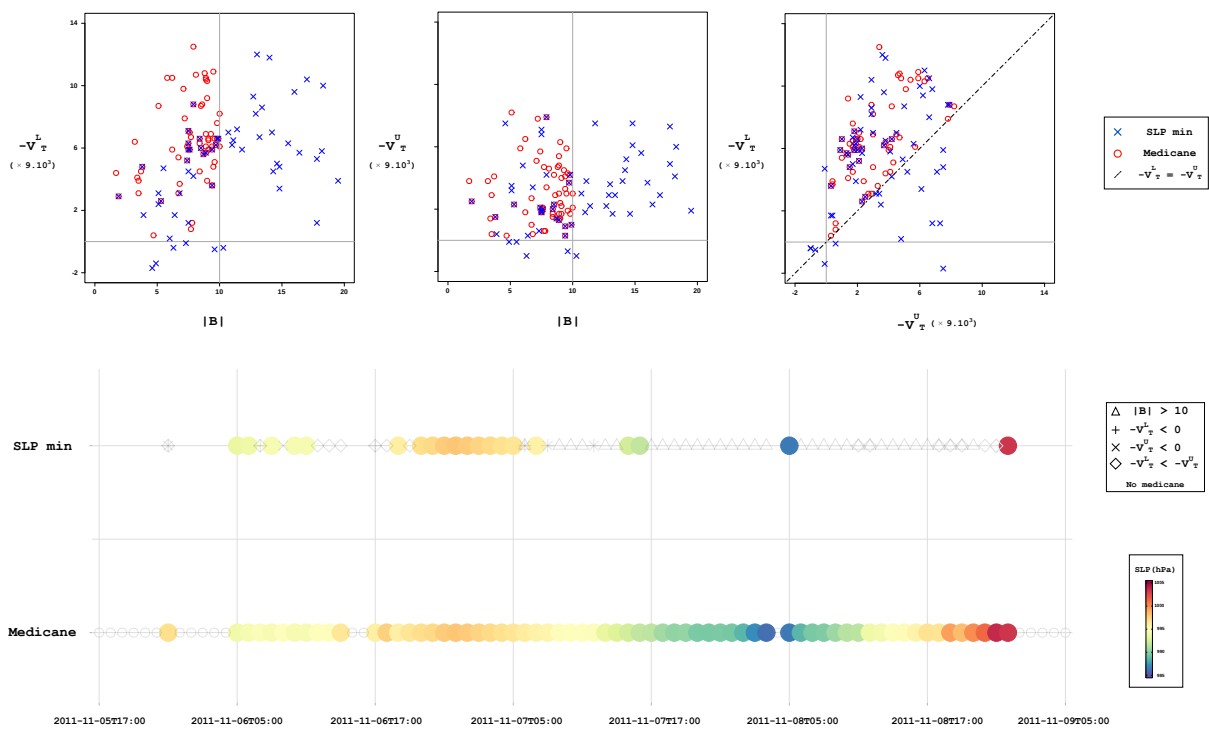

**Figure 3.** SLP minima and medicane centers for the Rolf medicane. In the top panel, the Hart phase space plots for points of SLP minimum (blue crosses) and centers detected by the algorithm (red circles). In the bottom plot, the temporal scheme of the detected centers and the SLP minimum track. Symbols indicate the Hart condition(s) not satisfied by the SLP minimum.

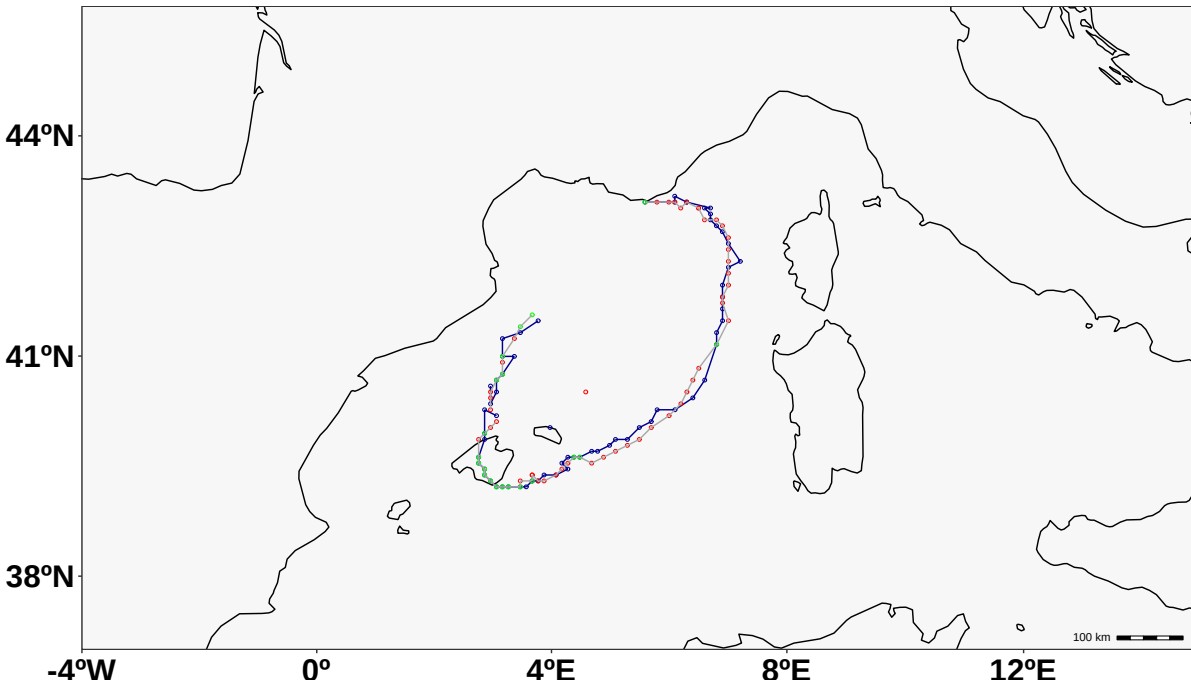

**Figure 4.** Rolf medicane tracks. Blue line represents the track calculated from the medicane centers found by the algorithm; grey line the one calculated with the SLP minimum. Green dots represent the points where the SLP minimum fulfills the Hart conditions and is selected as the medicane center. The red dots represent the SLP minimum when there is no coincidence with the medicane center detected by the algorithm (blue points).

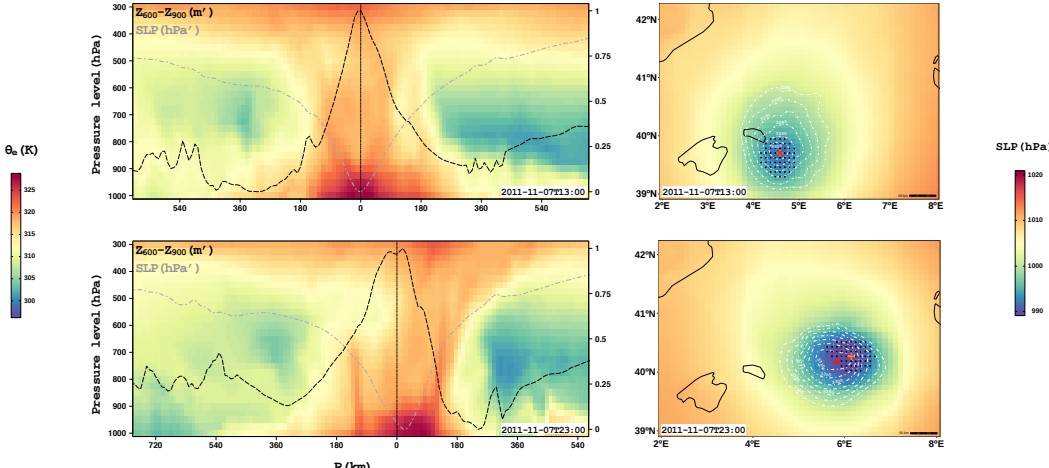

**Figure 5.** Depiction of the thermal structure of the Rolf medicane structure at two different time steps (top: 2011-11-07T16:00; bottom: 2011-11-07T23:00) by means of a zonal cross section (along the line of latitude passing through the medicane center found by the algorithm) of the equivalent potential temperature (colours on the left plots) and a contour plot of $Z_{600} - Z_{900}$ along with the SLP field in colours (right plots). In the left plots, the SLP (black dotted curve) and $Z_{600} - Z_{900}$ (grey dotted curve) are also presented, both scaled to the zero-one interval (unity-based normalization). A vertical line indicates the longitudinal position of the center found by the algorithm. In the right plots, dashed white lines show contours of the geopotential height thickness for the 900 hPa-600 hPa layer every 5 m starting from 3280 m. Additionally, the orange plus symbol specifies the position of the SLP minimum, while the red cross symbol denotes the position of the medicane center selected by the algorithm.

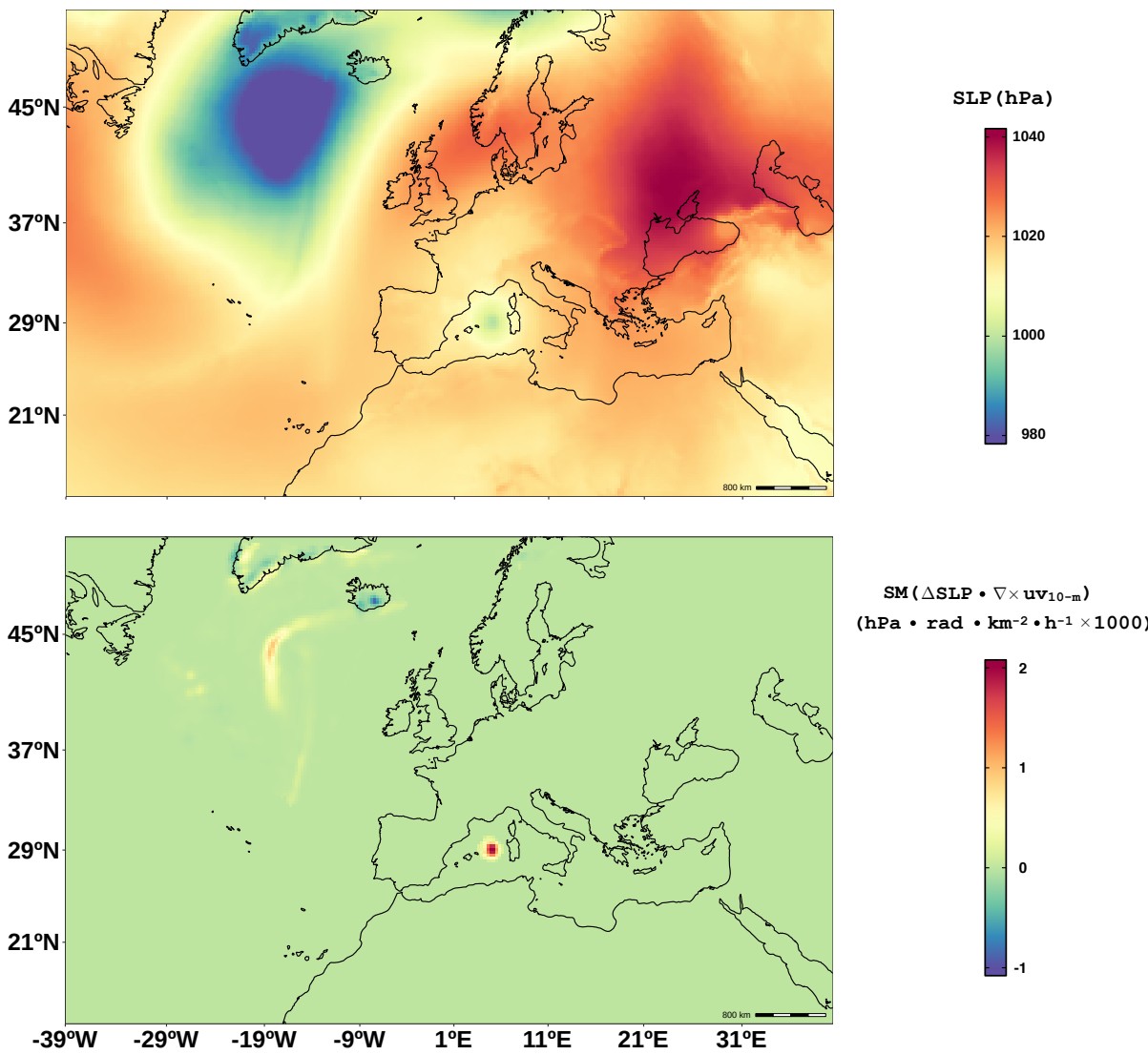

**Figure 6.** SLP (top) and scaled smoothed cyclonic potential $\mathcal{C}$ (bottom) for Rolf simulation at 27 km of grid spacing. Time step for both fields corresponds to 2011-11-07T12:00.

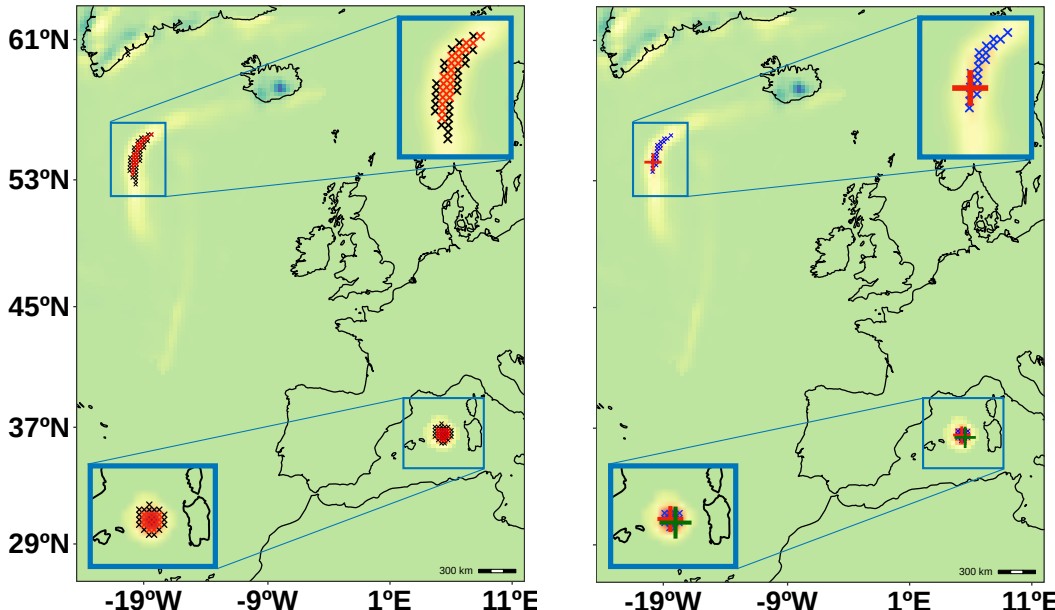

**Figure 7.** Scaled smoothed cyclonic potential $\mathcal{C}$ for Rolf simulation at 27 km of grid spacing at 2011-11-07T12:00, along with the points selected by the algorithm as potential medicane center candidates. Left: candidates after the quantile filter (black crosses) and after the vorticity threshold filter (red crosses). Right: candidates after the symmetry filter (blue crosses), cluster representative points (red plus symbols) and medicane center selected by the algorithm (green plus symbol).

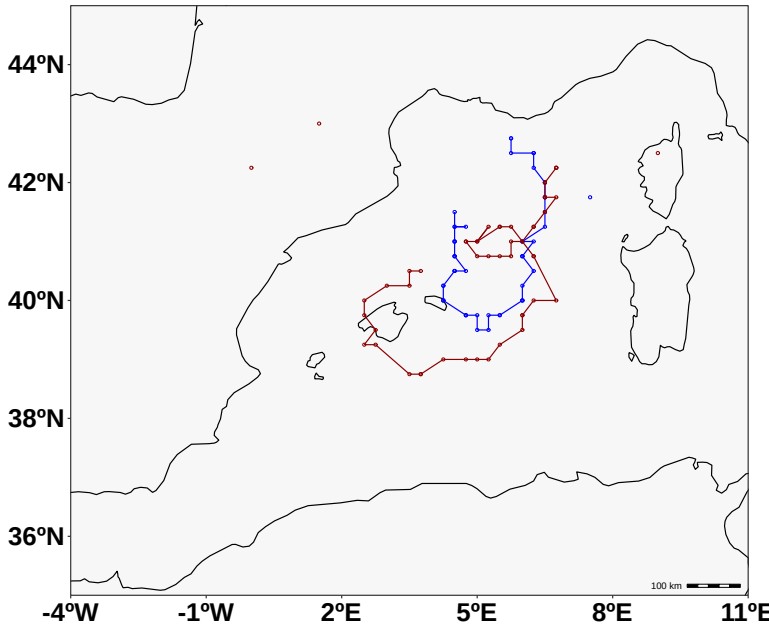

**Figure 8.** Rolf medicane tracking from WRF simulation at 27 km (blue track) and from ERA5 reanalysis data (dark red track) at 0.25º of grid spacing cropped to the Western Mediterranean area (see green box in Figure E1). ERA5 data is used in hourly resolution in order to get a precise track. The values of the algorithm parameters for these two simulations are the default ones, as indicated in Appendix A. However, for the B threshold, $B_{\text{threshold}} = 20$ m is used instead of 10 m provided that ERA5 shows a less intense medicane than the numerical simulations performed with WRF model, and thus the medicane structure is not so well defined, leading to a higher asymmetry. Changing the B parameter in the algorithm for the detection of medicanes in reanalysis data serves as a test for the algorithm flexibility and sensitivity to the different namelist parameters.

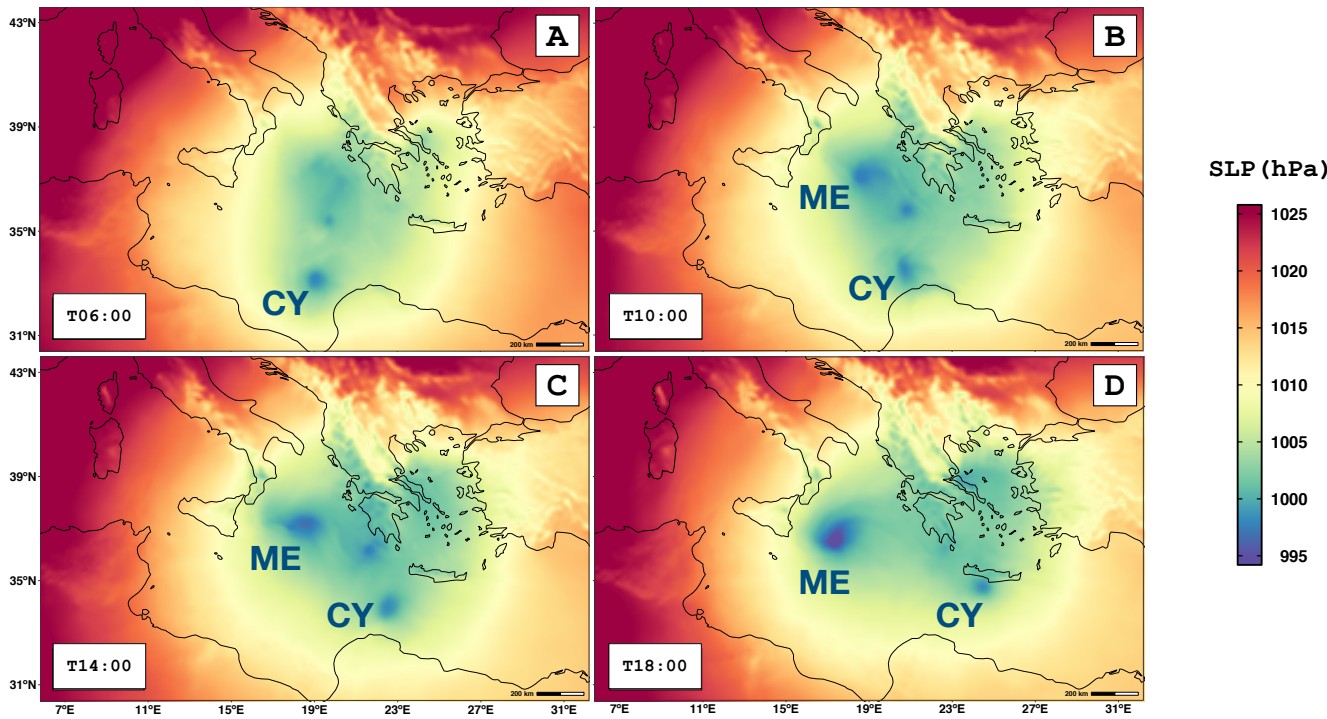

**Figure 9.** SLP field on 1995-01-14 at T06:00 (A), T10:00 (B), T14:00 (C) and T18:00 (D). The SLP minimum of the extratropical cyclone center is labeled with 'CY', while the medicane is marked with the 'ME' label.

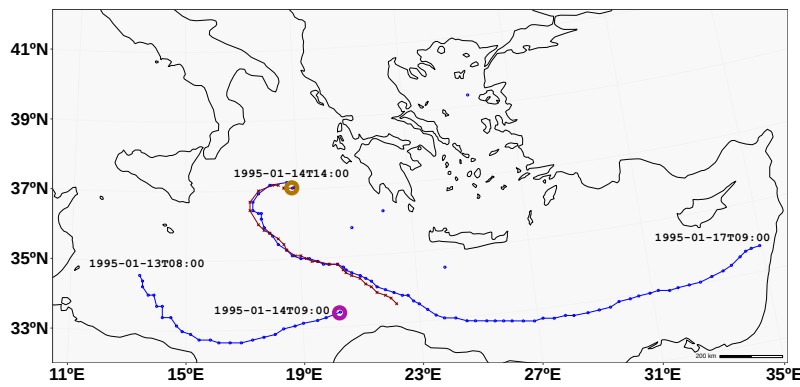

**Figure 10.** Tracks of cyclones between 1995-01-12 and 1995-01-18. Dark red line corresponds to the medicane track and blue lines represent the tracks calculated excluding the Hart conditions check in the algorithm namelist. Thus, blue lines are the complete cyclones track during their entire lifetime, while dark red line is the track of the cyclone when the conditions for being a medicane are fulfilled. The purple coloured circle represents the last point where an existing low-pressure center fulfills the filters (except the Hart conditions), while the gold one is the first location of another cyclone, which appears five hours after the extinction of the previous one and ends having a medicane structure (dark red line). The synoptic low (labelled as 'CY' in Figure 9) is not tracked from 1995-01-14T09:00 forward since it does not satisfy, among others, the symmetry condition.

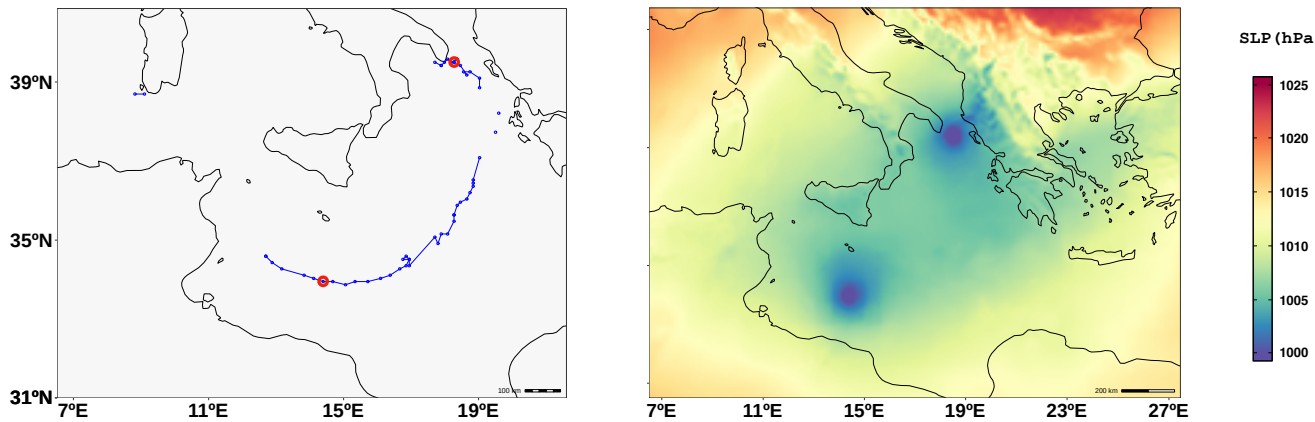

**Figure 11.** Tracks and SLP field for the case of Leucosia medicane simulation. Blue circles represent the medicane centers found in successive time steps (left panel). The two red coloured circles correspond to the location of the two medicanes at 1982-01-25T12:00. The right panel shows the SLP (hPa) for that time.

## Appendix A: Algorithm parameters description and default values

| Parameter | Definition | Default value |
|---|---|---|
| InitTime | Initial time step for the medicanes search. No medicanes will be found for timestamps before to this one. If string 'initial' is used, first timestamp in the input file will be used as initial time step. | initial |
| FinalTime | Final time step for the medicanes search. No medicanes will be found for timestamps after this one. If string 'final' is used, last timestamp in the input file will be used as initial time step. | final |
| Resolution | Spatial horizontal grid spacing of the netCDF (in km). Resolution is assumed to be the same in both directions. Future versions of the algorithm will support different grid spacings for both longitudinal and latitudinal dimensions for large grids in non-regular projections. It has no default value, so the string 'RR' is used and, if not changed, it will throw an error, as it is expecting a number. | RR |
| TimestepDt | Temporal resolution of the netCDF (in hours). Default value is 1 hour between netCDF timestamps. | 1 hour |
| LonDimName | Name of the longitude dimension in the netCDF. It takes the name 'west_east' for WRF-python output, and 'lon' for ERA5 and ERA-Interim reanalysis data. | west_east |
| LonVarName | Name of the longitude variable in the netCDF. It takes the name 'XLONG' for WRF-python output, and 'lon' for ERA5 and ERA-Interim reanalysis data. | XLONG |
| LatDimName | Name of the latitude dimension in the netCDF. It takes the name 'south_north' for WRF-python output, and 'lat' for ERA5 and ERA-Interim reanalysis data. | south_north |
| LatVarName | Name of the latitude variable in the netCDF. It takes the name 'XLAT' for WRF-python output, and 'lat' for ERA5 and ERA-Interim reanalysis data. | XLAT |

| | | |
|---|---|---|
| TimeDimName | Name of the time dimension in the netCDF. It takes the name 'Time' for WRF-python output, and 'time' for ERA5 and ERA-Interim reanalysis data. | Time |
| PressureVertLevelDimName | Name of the vertical levels dimension for 3D variables in the netCDF. It takes the name 'interp_level' for WRF-python output, and 'plev' for ERA5 and ERA-Interim reanalysis data. | interp_level |
| SLPVarName | Name of the SLP variable in the outputfile-slp.nc netCDF. It takes the name 'slp' for WRF-python output, and 'var151' for ERA5 and ERA-Interim reanalysis data. | slp |
| U10VarName | Name of the 10-m wind U variable in the outputfile-uvmet10-U.nc netCDF. It takes the name 'uvmet10' for WRF-python output, and 'var165' for ERA5 and ERA-Interim reanalysis data. | uvmet10 |
| V10VarName | Name of the 10-m wind V variable in the outputfile-uvmet10-V.nc netCDF. It takes the name 'uvmet10' for WRF-python output, and 'var166' for ERA5 and ERA-Interim reanalysis data. | uvmet10 |
| ZVarName | Name of the geopotential height variable in the outputfile-z.nc netCDF. It takes the name 'height' for WRF-python output, and 'var129' for ERA5 and ERA-Interim reanalysis data. | height |
| SmoothingPasses | Number of passes of the 1-2-1 smoothing of the product field. This product is the result of a point-wise multiplication of the SLP laplacian and the 10m wind rotational (vorticity at 10m -surface level-). The number of passes is the number of times that smoothing is sequentially performed. Default value is 5; a value above 3 is recommended. | 5 |
| SLPThreshold | Threshold for the first filter. It is a SLP minimum value, which should be fulfilled by every point being a center candidate. Defaults to 1005 hPa, which is expected to be exceeded on a medicane center. | 1005 hPa |

| | | |
|---|---|---|
| ProductQuantileLowerLimit | Parameter of the second filter. It represents the quantile lower limit applied to the product field, above which all points are selected as center candidates. This isn't a necessary filter from a physical view, but is a critical one for computational reasons. If not applied, we would have to calculate the Hart parameters for each grid point, which is highly expensive. Defaults to 0.999 (99.9 percentile). This means, in a 200x200 grid, only 40 points are selected as center candidates. | 0.999 |
| VorticityThreshold | Threshold for the third filter. It is a vorticity minimum value, which should be exceeded by every point being a center candidate. This filter is applied to the center candidates selected by the above quantile, and performs as an efficiency filter, avoiding the calculation of the Hart parameters in conditions of lack of vorticity in the domain, which is related with the absence of cyclonic activity. Defaults to $1 \text{ rad} \cdot \text{h}^{-1}$, a number obtained by means of our own ad hoc numerical study of vorticity typical values in the presence/absence of medicanes. | $1 \text{ rad} \cdot \text{h}^{-1}$ |
| CalculateZeroVortRadiusThreshold | Measure to calculate the variable radius which will be used in the calculation of Z gradient symmetry and Hart parameters. The options are 'zero' and 'mean'. If 'zero', the radius is calculated as the mean radial distance from the center to the zero vorticity line. If 'mean', to the contour of the vorticity mean domain value. Defaults to 'zero'. | zero |
| CalculateZeroVortRadiusDistance | Length of the lines along which vorticity sign change -if threshold is zero- or mean value -if threshold is mean- is searched in 8 directions. Determines the max size of the structures allowed in the domain, since if no critical point (zero or mean vorticity) is found on any of the directions, the point is discarded. Defaults to 300 km. | 300 km |

| | | |
|---|---|---|
| IfCheckZeroVortSymm | Whether to apply the zero vorticity symmetry filter, based on asking the contour of zero-vorticity around the center candidate to be axisymmetric. It is calculated taking eight directions and getting the distance at which the vorticity changes its sign. If this sign change is not reached in the number of points requested (see previous parameter), then it is set to Inf -1e10-. This filter stands on the fact that tropical cylones -and so, medicanes- must have a closed circulation . Defaults to TRUE. | TRUE |
| ZeroVortRadiusMaxAllowedAsymm | Maximum asymmetry (in km) allowed for the zero-vorticity radius calculation. This means that a center candidate is discarded if the difference between any pair of the eight calculated distances is higher than this allowed asymmetry value. The lower this parameter value is, the more restrictive is the symmetry condition imposed. Defaults to 300 km. | 300 km |
| ZeroVortRadiusMinSymmDirs | Minimum number of directions (out of 8) that should be non Inf. In other words, minimum number of directions in which a sign change should be found within the distance specified in the previous parameter. The higher the number of directions, the more is the symmetry requested. This prevents the method from failing in the cases of spiraling vorticity fields, where a large enough spiral arm matching the calculation direction could lead to constant signed vorticity values. Defaults to 6 directions (out of 8). | 6 |
| ZeroVortRadiusUpperLimit | Upper limit for the zero-vorticity radius. If a center candidate is calculated a zero-vorticity radius above this upper limit, it is discarded as a medicane center candidate. Medicane outer radius typical values are between 100 and 300 km. A non-restrictive default value of 1000 km is used. | 1000 km |

| | | |
|---|---|---|
| ZeroVortRadiusLowerLimit | Lower limit for the zero-vorticity radius. If a center candidate is calculated a zero-vorticity radius below this lower limit, it is discarded as a medicane center candidate. Medicane outer radius typical values are between 100 and 300 km. Default value is 80 km. | 80 km |
| SLPminsClustersMinIBdistance | The minimum distance between two points to be considered to belong to different clusters and, thus, to be candidates for two different medicane centers. This parameter should be directly related to the mean size of the cyclone that we are searching. Default value is 300 km, given that medicanes are usually between 100 and 200 km in radius. | 300 km |
| MaxNumberOfDifferentClusters | Maximum number of different cyclones that can be found in the analyzed domain at a given time step (i.e., the maximum allowed number of concurrent cyclones). If all restrictions are removed, the filters are ignored and the Hart conditions not checked, we would be searching cyclones, and in domains that are large enough, a huge amount of cyclones could appear. This is the motivation for the inclusion of this parameter. In case of being exceeded, the centers that will be found are the ones with higher product value, which means those with a greater cyclonic nature. Defaults to 50, a limit that is high enough when looking for medicanes and using all the filters, but could be surpassed for certain combinations of these parameters. | 50 |

| | | |
|---|---|---|
| MinPointsNumberInCluster | Filter to remove center candidates. Once that the center candidates are split into clusters that are farther than a certain distance from any other cluster, all the groups that contain less than a quantity of points are discarded. This number represents the minimum number of points that a group must have to be considered as a potential cyclone center. This is a filter oriented to remove orographic artifacts that, given their singular placement, can have high wind curl values and a positive value of the laplacian (interpolation effects may lead to artifacts in the slp surface, showing low values in orographic systems). However, these critical points are usually isolated, and hence removed with this filtering. Defaults to 5 points inside the cluster. Its value should be consistent with the number of points selected by the quantile filter. | 5 |
| IfCheckHartParamsConditions | The Hart parameters are three parameters stated by Hart in 2003 conceived to define in an objective manner the tropical nature of a cyclone. He defined a parameter B, directly related with the thermal symmetry of the cyclone, and two parameters of thermal wind in the lower and upper troposphere, showing a deep connection with the warm core nature of the system. From these three parameters, four conditions should be fulfilled by a tropical cyclone. Default value is TRUE and then Hart conditions are checked. | TRUE |
| HartConditionsTocheck | The Hart conditions are: 1. $B < B_{threshold}$ -m- (see parameter $B_{threshold}$); 2. $-V_T^L > 0$; 3. $-V_T^U > 0$; 4. $-V_T^L > -V_T^U$. If Hart conditions are checked -i.e., previous parameter is set to TRUE- any condition can be removed and won't be necessarily TRUE for a point to be considered a medicane. Defaults to 1,2,3,4 and all the conditions are checked. | 1,2,3,4 |
| Blowerpressurelevel | Lower pressure level for the calculation of the B parameter (Hart 2003). Defaults to 900 hPa. | 900 hPa |

| | | |
|---|---|---|
| Bupperpressurelevel | Upper pressure level for the calculation of the B parameter. Defaults to 600 hPa. | 600 hPa |
| Bmultiplemeasure | If multiple directions are used to calculate a more constrained B parameter, the measure to use. Defaults to 'max' | max |
| Bdirections | Number of directions to be used in the calculation of the more restrictive B parameter. The maximum allowed is 4 directions, and at least 2 directions are recommended. Defaults to 4 directions. | 4 |
| Bthreshold | Threshold -in meters- of thermal symmetry parameter B. It represents the maximum allowed thermal asymmetry in the thickness of the geopotential height layer between the left and the right side of a circle centered in the point checked, and divided by a vector in the direction of motion of the cyclone. Hart recommends a value of 10 meters for tropical cyclones. Although this may be a too strong limitation for medicanes, whose symmetry is not as well defined as in the former ones, a default value of 10 meters is used for the threshold of B. | 10 m |
| LTWlowerpressurelevel | Lower pressure level for the calculation of the $V_T^L$ (lower tropospheric thermal wind) parameter. Defaults to 900 hPa. | 900 hPa |
| LTWupperpressurelevel | Upper pressure level for the calculation of the $V_T^L$ (lower tropospheric thermal wind) parameter. Defaults to 600 hPa. | 600 hPa |
| UTWlowerpressurelevel | Lower pressure level for the calculation of the $V_T^U$ (upper tropospheric thermal wind) parameter. Defaults to 600 hPa. | 600 hPa |
| UTWupperpressurelevel | Upper pressure level for the calculation of the $V_T^U$ (upper tropospheric thermal wind) parameter. Defaults to 300 hPa. | 300 hPa |

## Appendix B: Algorithm input specifications

As mentioned in Section 2, the input data of the algorithm described in this paper consist of multiple netCDF (.nc) files containing temporal series of certain meteorological fields. The mandatory 2D and 3D fields are Sea Level Pressure (SLP), 10-m wind horizontal components (U10, V10) and geopotential height (Z) for, at least, the 900, 800, 700, 600, 500, 400 and 300 hPa levels. Note that the more vertical levels, the more precise will be the Hart thermal wind parameters calculation (a minimum of 20 vertical levels is recommended for obtaining trustworthy results). The requested units for the fields are hPa for SLP, meters for geopotential height, and $km \cdot h^{-1}$ for both 10-m wind horizontal components.

If a WRF output file is to be used as input data for the algorithm, then the use of the provided pinterpy package is strongly recommended (which can be found at the GitHub repository github:eps22/TITAM). In the namelist file 'interp-namelist', the input file name must be changed to the WRF output file containing all the time steps (*ncrcat* command of NCO tools is referred for the task of temporal merge). Detailed instructions on the requested python version and libraries for a successful running can be found at github:eps22/TITAM/README.md, while specific pinterpy usage instructions and a detailed description of the namelist parameters can be found at github:eps22/TITAM/Code/pinterpy/README.interp-namelist.

In case of using input data different from WRF output, the metadata must be closely inspected and the following parameters must be set accordingly in the FindMedicanes.namelist file: *LonDimName, LonVarName, LatDimName, LatVarName, TimeDimName, PressureVertLevelDimName, SLPVarName, U10VarName, V10VarName* and *ZVarName*. The vertical levels in the geopotential height 3D field do not need to follow a specific order, and both increasing and decreasing sortings are allowed and automatically detected.

## Appendix C: Technical notes on the algorithm deployment and multi-core performance

The algorithm execution requires prior installation of the R environment with the 'ncdf4' and 'oce' libraries. Details on the recommended R version and the 'oce' library installation process can be found at github:eps22/TITAM/README.md.

As mentioned in Sections 1 and 3, multi-core parallel computing is supported and encouraged. The libraries foreach and doParallel are requested for this type of execution. If these libraries are not installed or will not be required (single core run), the flag for the number of cores, 1, needs to be used as second argument when running the algorithm, being the first argument the input file or folder. See further details at github:eps22/TITAM/README.md.

Regarding the parallelization implemented in the algorithm, we test its performance by means of different algorithm executions over the Rolf simulation with 27 km of grid spacing (described in Appendix E and analyzed in Section 4.2). The left plot of Figure C1 shows the execution times for the different runs of the algorithm changing the number of processors for the calculation (black dots).

In computer science, Amdahl's law (Amdahl, 1967) defines the speedup achieved when increasing the number of processors that compute in parallel as a function of the proportion of the code that must be processed serially ($P$). It is often expressed as:

$$\mathcal{S} = \frac{1}{P + \frac{1-P}{N}} \tag{C1}$$

where $\mathcal{S}$ is the speedup, $P$ the non-parallelizable proportion of code, and N the number of processors. In the particular case of a fully parallelizable code ($P$=0), there is a 'linear speedup' when increasing the number of processors (blue line in the left plot of Figure C1). In this same plot, an adjust of a theoretical curve (red line) following Amdahl's law to our execution times (black dots) shows that for the particular case of the Rolf simulation at 27 km, $P$=0.087, which means that 91.5% of the code is run in parallel. The right plot of Figure C1 shows the theoretical speedup curve obeying Amdahl's law for $P$=0.087 with increasing number of processors, reaching an asymptote at $\mathcal{S} = 1/0.087 \simeq 11.5$ for $N \to \infty$.

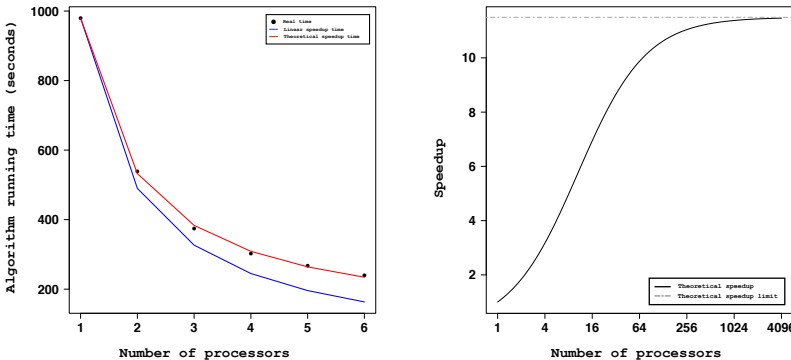

**Figure C1.** Parallel performance of the tracking algorithm. To the left, the experimental execution times of the algorithm (black dots) as a function of the number of processors for the parallel computing. The red curve represents the fit of the Amdahl's law to the data ($P$=0.087); in blue, the 'linear speedup' theoretical curve ($P$=0). To the right, the adjusted Amdahl's law curve ($P$=0.087) versus the number of processors (solid black line), asymptotically reaching $\mathcal{S} = 11.5$ (dashed grey line).

## Appendix D: Postprocessing tools included in the package

An additional tool is provided to extract further information on the medicane size and intensity. Provided the RData file, output of the medicanes tracking algorithm, the *getmedicanestrackdata* bash script diagnoses additional variables from the found medicane centers. In the 'reduced' mode, only longitude, latitude and SLP value of the medicane center are calculated. The 'complete' method extends to other variables, such as the minimum SLP value inside the zero-vorticity radius and its position, or the 10-m maximum wind speed inside the medicane domain, which allows the classification of the medicane category in terms of its intensity as defined in the Saffir-Simpson scale. Detailed information about this postprocessing tool can be found at github:eps22/TITAM/README.md.

Moreover, in Section 3.4 we defined the rules to connect two found medicane centers. Once the isolated points are connected, our next step is to create a plot with the calculated medicane track. To this end, an auxiliar plotting script is provided (see github:eps22/TITAM/README.md for detailed instructions on its usage). Based on a more generic plotting function (github:eps22/TITAM/Code/PostProcessing/MatrixPlot.R), the 'plotmedicanestrack' bash script produces a pdf receiving an RData file (output of the tracking algorithm) and the netCDF files as input data.

It is also important to highlight that the function to plot the calculated medicane tracking expects either a regular grid in lon-lat projection or an irregular one in a Lambert projection. Please note that this postprocessing tool is not prepared to receive input data expressed in any other projection, although the tracking algorithm will run successfully. If the input data is neither WRF output nor lon-lat projected data, lines 49 to 62 of TITAM/Code/PostProcessing/PlotTrack.R must be commented out and CRS (Coordinate Reference System) must be set in proj4string notation according to the projection of the data, in order to get an output map properly projected.

## Appendix E: Review of the utilized WRF simulations

Given the relatively small horizontal extent of medicanes, fine grid spacing fields are needed to correctly interpret their thermal properties and dynamics. To achieve this high resolution, dynamical downscaling is often employed by means of the so called RCMs (Regional Climate Models). For this study we produce the necessary meteorological fields for initial and boundary conditions by downscaling the ERA-Interim reanalysis with the WRF Model (Skamarock et al., 2008). This model is highly sensible to the domain configuration and set of parameterizations that determine how the dynamics, physical and chemical mechanisms (in the case of the WRF-chem coupled model) are solved. However, given that this work focuses on the algorithm, rather than on the ability of the model to accurately reproduce medicane characteristics, we have kept fixed the model configuration to one that is physically consistent with the medicane main features and fostering processes.

No physics suite (WRF preconfiguration of a set of well-tested physics parameterizations as a suite) is used for the model run. The chosen parameterizations lead to the following physical configurations of the model: the Morrison et al. (2009) second-moment microphysical scheme is used (*mp_physics=10*) and prognostic cloud droplet number is included in the Morrison microphysics scheme (*progn*=1). Radiation is parameterized with the Rapid Radiative Transfer Model for GCMs (RRTMG) by Mlawer et al. (1997), both for short and long wave radiation, solved each 30 minutes. Additionally, the selected option for the surface layer parameterization solves with the MM5 scheme based on the similarity theory by Monin and Obukhov (1954), while the *Unified NOAH LSM* option is used for the land-surface calculation (Mitchell, 2005). The number of soil layers in land surface model is thus 4. Yonsei University scheme is employed for the boundary layer (Hong et al., 2006), solved every time step (*bldt*=0). For the cumulus physics, Grell 3D ensemble (*cu_physics*=5; *cudt*=0) is chosen to parameterize convection (Grell and Dévényi, 2002). Heat and moisture fluxes from the surface are activated (*isfflx*=1), as well as the cloud effect to the optical depth in radiation (*icloud*=1). Conversely, snow-cover effects are deactivated (*ifsnow*=0). Landuse and soil category data come from WPSgeogrid but with dominant categories recomputed (*surface_input_source* = 1). Urban canopy model is not considered (*sf_urban_physics*=0), and the topographic surface wind correction from Jiménez and Dudhia (2012) is turned

on. Both feedback from the parameterized convection to the radiation schemes and SST update (every 6 hours, coinciding with boundary conditions update) are also turned on.

As exposed throughout the text, we have selected a number of historical events that cover a range of structures which serve as a testbed for the description and evaluation of the tracking algorithm. In particular, four different events have been simulated for the sake of the algorithm testing:

– Simulation of Rolf medicane with 9 km of grid spacing. This event spans the period from 2011-11-05 to 2011-11-10 with hourly resolution. This 9 km inner domain (blue bounding box in Figure E1) is nested to a larger domain, which includes the Iberian Peninsula, Balearic Islands and the territory of Italy with a coarser grid spacing of 27 km. The large domain is run with spectral nudging to ERA-Interim global data for wavelengths above 1000 km.

– Simulation of Rolf medicane with 27 km of grid spacing and hourly temporal resolution. As the previous case, it includes the time range from 2011-11-05 to 2011-11-10. A single large domain of 27 km is included, which covers the East Atlantic and Mediterranean areas, and latitudes from the north of Africa to Greenland (red bounding box in Figure E1). This domain is also run with spectral nudging to ERA-Interim global data for wavelengths above 1000 km.

– Simulation of Celeno medicane between 1995-01-12 and 1995-01-18 run with 9 km of grid spacing and hourly temporal resolution. This simulation is based on a first large domain of 27 km including all the Mediterranean basin with a nested domain of 9 km covering the eastern Mediterranean area (orange bounding box in Figure E1). The coarser resolution domain is run with spectral nudging to ERA-Interim global data for wavelengths above 1000 km.

– Simulation of Leucosia medicane from 1982-01-19 to 1982-01-28 with hourly resolution in a small domain with 9 km of grid spacing (orange bounding box in Figure E1) nested to a larger 27 km domain limited to the Mediterranean area. The large domain is run with spectral nudging to ERA-Interim global data for wavelengths above 1000 km.

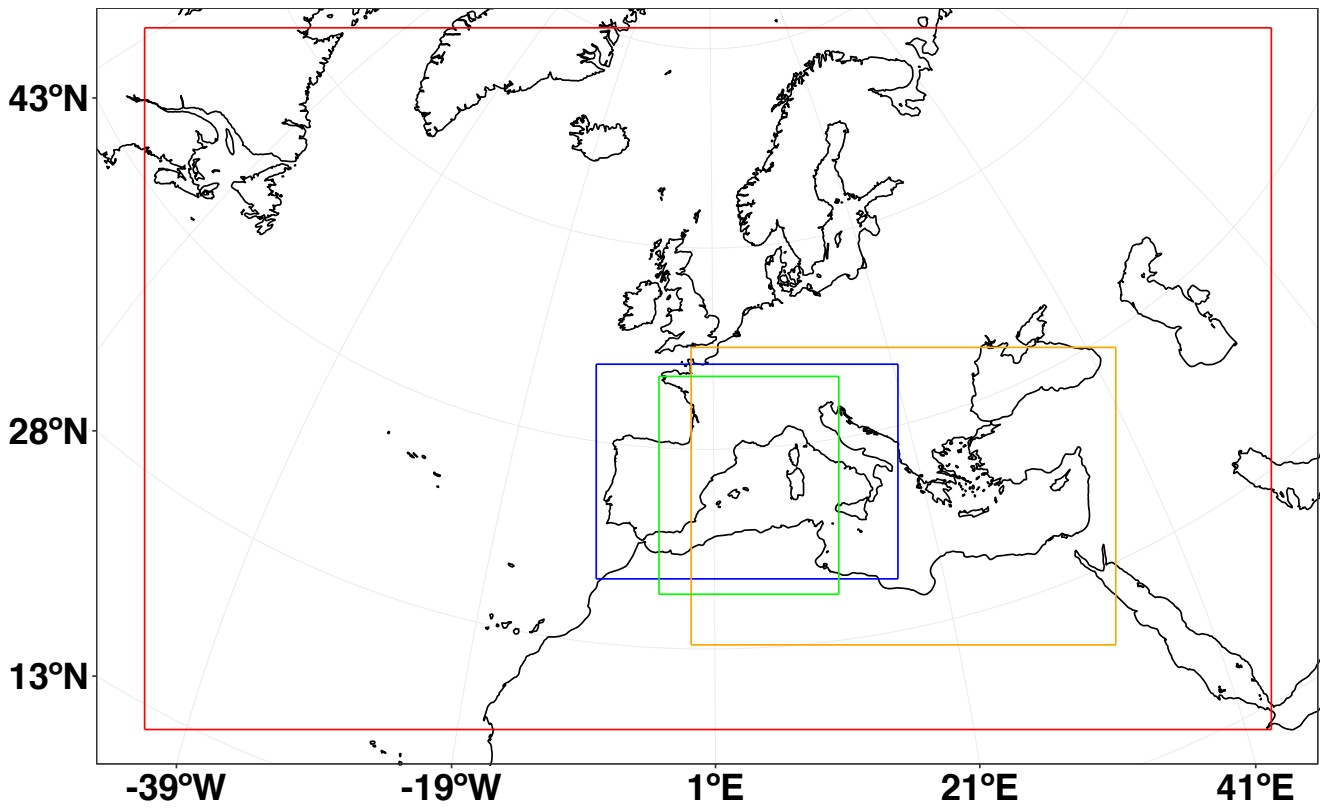

**Figure E1.** Spatial domains covered by the WRF simulations described above. Domains correspond to the following simulations: Rolf at 27 km (red), Rolf at 9 km (blue), Celeno and Leucosia at 9 km (orange). Additionally, the green box covers the spatial area selected to run the algorithm on ERA5 data.

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
