# Peer review of "TITAM (v1.0): Time Independent Tracking Algorithm for Medicanes"

_Geoscientific Model Development, 2020_

## Referee Comment (RC1) · Anonymous Referee #1 · 14 Jul 2020

Title: TITAM (v1.0): Time Independent Tracking Algorithm for Medicanes Authors: Pravia-Sarabia et al. RECOMMENDATION: Major revisions

General comments Medicanes are receiving a growing attention in literature and among stakeholders due to their potential damage to coastal zones. In this framework, the Authors of the present paper provide a new algorithm for the detection and tracking of these cyclones, following an innovative, time-independent approach. The paper is interesting not only for the development of the new software, but also because the adopted procedure sheds some new lights on the structure of medicanes, allowing a better understanding of their properties. The paper appears an interesting contribution in the growing literature in the field, however some major revisions are needed before publication.

[Figure]

Specific comments - While the proposed algorithm has been well designed, and takes into account all the properties of this category of cyclones, I was wondering if it allows to track the whole cyclone lifetime as a unique track (not different tracks for different stages of the cyclone lifetime, as in Fig. 10). This is an important point, also considering that the most intense convection is often observed in this earlier stage (Dafis et al., 2018; Miglietta et al., 2013). - The description of the mechanisms of development of Medicanes is poor and confusing. I recommend the Authors to completely re-write this section, starting from the explanation in Miglietta and Rotunno (2019) and related bibliography. In particular: L22: they do not tend to acquire a cold core, they always start with a cold core; L24: Really in WISHE theory for tropical cyclone development (Emanuel, 1986), the role of cumulus convection is to redistribute the heat acquired from sea surface, so storms result from an air-sea interaction instability. Please, clarify your sentence in the framework of tropical cyclone theory; L31: the term genesis is not appropriate since they evolve from extra-tropical cyclones, so they are already formed before they acquire tropical features; L32: "These storms are very close to a tropical cyclone on its fundamentals": really, Miglietta and Rotunno (2019) clarified better the similarity of Medicanes with tropical cyclones for their dynamical and thermodynamic properties, which is somewhat case dependent; L33: "as well as on the trigger mechanisms and necessary conditions for its genesis": please explain the differences with tropical cyclones; L34: "similar mechanisms": again, please take into consideration the results in Miglietta and Rotunno (2019); L37-38: the sentence is quite confusing, since the environment you describe is unstable, not stable as you state; L39-40: this is valid also in tropical areas, since a part of tropical cyclones develops in a way similar to Medicanes, with SST below the classical threshold of 26°C (McTaggart-Cowen et al., 2015); L41-42: "Once the vertical moist air fluxes appear, advection takes place and the core starts heating due to the latent heat release": this sequence is not clear, since advection already occurs in a baroclinic environment even in the absence of fluxes; L42-43: "The development of a warm core system then leads to an axi-symmetric storm by means of the cyclonic rotation of air around the center, induced by advection

of relative vorticity towards the low pressure center": this sentence is very confusing: why do warm core and cyclonic rotation imply axi-symmetric storm? is cyclonic rotation induced by advection of relative vorticity? L49: see also Table 1 in Miglietta et al. (2013) - The other algorithms in the literature, which have mentioned in the Introduction, should be described with further details, in order to appreciate the innovative features of the proposed algorithm; possibly, the characteristics of the different methods could be summarized in a Table; also, the explanation of this part should be improved, in particular: L57-58: "the existence of two different low pressure areas is equivalent to the existence of two medicanes": do you mean that identifying two cyclones is a similar problem as the identification of two medicanes? L65-66: "even in the absence of an optimal medicane definition, the detection would be ensured within a reasonable range of the parameters leading to that definition": please clarify; L73: in what does the methodology introduced in Alpert et al. (1990) consist? L76-77: what do you mean with "points fulfilling the Hart parameters"? L77-78: what do you mean with "large gaps could be observed in the tropical cyclonic nature of the calculated tracks"? L82-83: delete "This approach by Hart (2003) consists in a track identification by imposing a series of conditions to spatial displacement of two time consecutive medicane centers", since it repeats the previous concept;

Minor and Technical corrections L5: the former ones: the sentence is not clear, since "the former" does not refer to anything. L6: temporarily lose . . .: really, since they appear in baroclinic environments, they temporarily lose their cold-cored and asymmetric structure; L33: their -> its L43, L65: In this way, . . . L135: rotation -> rotational L169: quasi -> quasy L196: requirements -> requeriments L203-L205: please clarify the sentence L244: is there a motivation for imposing - | VTL | > - | VTU |? L273: conversely, if the condition is valid, do you connect the positions at different i? L288: the same case is also discussed in Dafis et al. (2018) and Ricchi et al. (2017) L289: . . . and long lifetime . . . L291 and elsewhere: grid spacing not horizontal resolution, the two concepts are different (see Skamarock, 2004). L299: I cannot identify the black crosses in Fig. 2 bottom panel L328: 17H: what does H stand for? L347: please use

the same format for the time in the whole manuscript L349: Atlantic -> Altantic L355: green plus for Medicanes L357: successfully -> succesfully L369: see also Lagouvardos et al. (1999) L385: see Ernst and Mason (1983), Reed et al. (2001) L402: what do you mean with "methodology which does not get trapped in previous perturbations"? Figure 5 caption: what do you mean with normalized? Figure 8 caption: color line meaning is missing; why using a different threshold for B? Figure 11 caption: year not indicated Appendix A: ERA-Interim not ERA5-Interim Appendix A: "a number obtained by means of a numerical study of vorticity typical values in the presence/absence of medicanes": do you refer to a published paper? Appendix A: "ZeroVortRadiusLowerLimit": occasionally Medicanes can be smaller than 80 km (see Miglietta et al., 2013) Appendix A: In "MinPointsNumberInCluster" farther instead of further L505: what does CRS mean? L512: WRF does not include chemical mechanisms, WRF-CHEM does. L516: what do you mean with "No physics suite is used for the model run"? Figure E1: "Additionally, the green box covers the spatial area selected to run the algorithm on ERA5 data": is this for all Medicanes?

BIBLIOGRAPHY: Dafis, S., Rysman, J.-F., Claud, C. and Flaounas, E. (2018) Remote sensing of deep convection within a tropical-like cyclone over the Mediterranean Sea. Atmospheric Science Letters, 19(6), e823. Emanuel, K.A. An air-sea interaction theory for tropical cyclones. Part I: Steady-state maintenance. J. Atmos. Sci. 1986, 43, 585–604. Ernst, J. A., and M. Matson, 1983: A Mediterranean tropical storm? Weather, 38, 332–337. Lagouvardos, K., Kotroni, V., Nickovic, S., Jovic, D. and Kallos, G. (1999) Observations and model simulations of a winter sub-synoptic vortex over the central Mediterranean. Meteorological Applications, 6, 371–383. McTaggart-Cowan, R., Davies, E.L., Fairman, J.G., Jr., Galarneau, T.J., Jr. and Schultz, D.M. (2015) Revisiting the 26.5ŮęC sea surface temperature threshold for tropical cyclone development. Bulletin of the American Meteorological Society, 96, 1929–1943. Miglietta, M.M., Laviola, S., Malvaldi, A., Conte, D., Levizzani, V. and Price, C. (2013) Analysis of tropical-like cyclones over the Mediterranean Sea through a combined modelling and satellite approach. Geophysical Research Letters, 40, 2400–2405. Miglietta, M.M.; Rotunno, R. Development Mechanisms for Mediterranean Tropical-Like Cyclones (Medicanes). Q. J. R. Meteorol. Soc. 2019. Reed, R.J., Kuo, Y.-H., Albright, M.D., Gao, K., Guo, Y.-R. and Huang,W. (2001) Analysis and modeling of a tropical-like cyclone in the Mediterranean Sea. Meteorology and Atmospheric Physics, 76, 183–202. Ricchi, A., Miglietta, M.M., Barbariol, F., Benetazzo, A., Bergamasco, A., Bonaldo, D., Cassardo, C., Falcieri, F.M.,Modugno, G., Russo, A., Sclavo,M. and Carniel, S. (2017) Sensitivity of a Mediterranean tropical-like cyclone to different model configurations and coupling strategies. Atmosphere, 8(5), 92, 1–32. https://doi.org/10.3390/atmos8050092. Skamarock W. C., Evaluating Mesoscale NWP Models Using Kinetic Energy Spectra, Mon. Wea. Rev. (2004) 132 (12): 3019–3032.

---

## Referee Comment (RC2) · Anonymous Referee #2 · 22 Aug 2020

**Summary**

The article presents a set of criteria designed to detect mesoscale cyclones in the Mediterranean Sea and surrounding areas that have characteristics similar to barotropic tropical cyclones despite being located at latitudes where baroclinic cyclones are most common. Successful demonstrations are presented for a few selected datasets that are known to contain these "medicane" cyclone features. The results are also shown in the "phase space" model advocated by Hart (2003).

**General comments**

The greatest contribution of this article are the criteria it uses to distinguish medicanes from typical extratropical cyclones. The use of the "cyclonic potential," and its motivation via connection to quasi-geostrophic theory is (to my knowledge) novel and a strength of the article. These criteria are actually independent of the algorithm and software employed by the authors which, as the title suggests, is a main focus of the article.

Unfortunately, the software and algorithms advocated by the article are not new. Indeed, the algorithm is the same as the commonly used 2-step search that is described in greater detail by Bosler et.al. (2016), Ullrich  Zarzycki (2017), Wernli  Schweirtz (2006), and Zhao et. al. (2009), and each of these references succeed previous implementations of the same basic algorithm (e.g., Blender et.al. (1997), Hodges (1994), and Vitart et.al (1997) that are themselves successors of even earlier work. Even this list of references is, therefore, far from complete, and more specialized studies using this algorithm that are also relevant to this work include Hanley  Caballero (2012), which (like the present work) addresses multiple circulations within the same larger system. None of the references in this review are cited by the current article, which is a significant omission, as it is not clear how (if at all) the present work distinguishes itself from them.

As a consequence, I cannot recommend this article for publication.

However, I would like to encourage the authors to reexamine their work from the context of their specific search criteria, which appear to be very successful at identifying medicanes. These methods could be applied to a larger dataset to examine the climatology of such storms, as in Zhao et. al (2009); the sensitivity of these climatologies could be examined with respect to different threshold values or criteria choices, as in Horn et. al. (2014). I also commend the authors for employing the cyclone "phase space" model as an analysis tool, and encourage them to continue to use it as this work matures. The focus in this ongoing work should not be on software. While writing software is undoubtedly where the authors spend much of their time and effort, this is the nature of modern science. The algorithms and code accompanying this article is not novel; it is simply one of many software packages that have been developed for

similar applications in recent (and not-so recent) years. Instead, the application the authors have chosen is an excellent topic for additional study, particularly as resolution (both model and data set) increases to the point that medicanes are well-resolved.

**Specific comments**

1. Figure 2(a): The legend label (SLP) is incorrect; the field shown is $\nabla^2(SLP)$.

2. Line 298 reports that detected storms are shown in Figure 2(c). These are not visible in my .pdf copy. Also, it appears that there are many (56?) such detections — why? How does this number correspond to the number of time steps in the data set?

3. Line 305: What is gained by not using the SLP minimum as the location of the storm? In figure 4, the SLP minimum produces a clearly smoother track.

4. Line 324: In what sense is the current method more "robust" than one that uses all the same criteria but chooses to define the location of the storm as the SLP minimum? This is one of many unquantifiable comments in the article that are better characterized as "sales" than scientific analysis.

5. For a study intended to identify medicanes, the ability to distinguish a North Atlantic storm seems (as shown in Figure 7) seems irrelevant. A better example would be an extratropical cyclone, associated with a digging trough, in one part of the Mediterannean basin and a separate system elsewhere in the region that contained a medicane, presuming such a situation can be found or simulated.

**References**

R. Blender, K. Fraedrich, and F. Lunkeit, 1997, Identification of cyclone-track regimes in the North Atlantic, *Q. J. Roy. Meteor. Soc.*, 123:727–741.

P.A. Bosler, E.L. Roesler, M.A. Taylor, and M.R. Mundt, 2016, Stride Search: a general algorithm for storm detection in high-resolution climate data, *Geosci. Model Dev.* 9:1383–1398.

J. Hanley and R. Caballero, 2012, Objective identification and tracking of multicentre cyclones in the ERA-Interim reanalaysis dataset, *Q. J. Roy. Meteor. Soc.* 138:612–625.

M. Horn et. al., 2014, Tracking scheme dependence of simulated tropical cyclone response to idealized climate simulations, *J. Climate* 27:9197–9213.

K.I. Hodges, 1994, A general method for tracking analysis and its application to meteorological data, *Mon. Weather Rev.* 122:2573–2586.

P.A. Ullrich and C.M. Zarzycki, 2017, TempestExtremes: a framework for scale-insensitive pointwise feature tracking on unstructured grids, *Geosci. Model Dev.* 10:1069–1090.

F. Vitart, J. L. Anderson, and W.F. Stern, 1997, Simulation of interannual variability of tropical storm frequency in an ensemble of GCM integrations, *J. Climate*, 10, 745–760.

H. Wernli and C. Schwierz, Surface cyclones in the ERA-40 dataset (1958–2001). Part I: Novel identification method and global climatology, *J. Atmos. Sci.*, 63, 2486–2507.

M. Zhao, I. M. Held, S.J. Lin, and G.A. Vecchi, 2009, Simulations of global hurricane climatology, interannual variability, and response to global warming using a 50-km resolution GCM, *J. Climate* 22(24):6653–6678.

---

## Author Comment (AC1) · 15 Sep 2020

[1]EnriquePravia-Sarabia  [1]Juan  JoséGómez-Navarro  [1,2]PedroJiménez-Guerrero
[1]Juan PedroMontávez

[1]Physics of the Earth, Regional Campus of International Excellence "Campus Mare
Nostrum", University of Murcia, 30100 Murcia, Spain [2]Biomedical Research Institute
of Murcia (IMIB-Arrixaca), 30120 Murcia, Spain

11

[Figure]

**TITAM (v1.0): Time Independent Tracking Algorithm for Medicanes. Reply to RC1.**

September 15, 2020

Referee comments: in blue.

Author responses: in black.

We welcome the feedback and appreciate all the referee suggestions and comments. The majority of them have been included in the final manuscript version.

Related to the general comments, thanks for your kind words.

Now we proceed with the responses to the specific comments. Please note that the changes associated to these comments can be found in the changes version generated with the latexdiff tool.

While the proposed algorithm has been well designed, and takes into account all the properties of this category of cyclones, I was wondering if it allows to track the whole cyclone lifetime as a unique track (not different tracks for different stages of the cyclone lifetime, as in Fig. 10). This is an important point, also considering that the most intense convection is often observed in this earlier stage (Dafis et al., 2018; Miglietta et al., 2013).

The algorithm is indeed designed to produce a single medicane track. If a cyclone tracking is expected to be the product of the algorithm execution, then the namelist parameters should be changed accordingly. Figure 10 is an example of a specific situation that could be problematic for a tracking procedure. On it, a large cyclone encompasses a SLP minimum and a distant zone where a medicane is formed. Our intention with the inclusion of Figure 10 was to show that if namelist options are properly adapted, a complete cyclone track can be found. In the same Figure 10, red track shows the path of the medicane, while the blue line represents the cyclone track. Disconnection between both blue paths only means that the algorithm is able to 'jump' from the SLP minimum of the large low pressure air mass, seen as a normal cyclone before the appearance of the medicane, to the medicane (please see Figure 9 for the evolution of the synoptic situation), while the SLP cyclone below loses the structure and is not followed by the algorithm anymore. Please note that in case the cyclone in the North coast of Libya did not lose its structure, both the medicane and the cyclone would be tracked simultaneously. Their tracks would also be disconnected, since they are not part of the same cyclone, even when they are formed from the same low pressure air mass. Thus, though in this particular case the medicane appears disconnected from the cyclone in the early stage, the different lengths of the blue and red tracks of the medicane show that the algorithm is able to track a cyclone as a unique track during its entire lifetime, regardless of whether it fulfills the conditions to be a medicane or not. It seems also important to mention that if Hart conditions are not checked in order to find the complete cyclone track, it will be necessary a second and independent algorithm execution to know the points in which the cyclone is a medicane, provided that the algorithm is not prepared to simultaneously provide a complete track and the points of the complete track where the cyclone shows a medicane structure.

However, for the sake of clarity, we have modified the explanation on Figure 10.

The description of the mechanisms of development of Medicanes is poor and confusing. I recommend the Authors to completely re-write this section, starting from the

explanation in Miglietta and Rotunno (2019) and related bibliography.

As suggested, we have completely re-written this section. Thanks for the constructive comments. For simplicity, we skip the comments in which we have directly considered the suggestions, and will only reply to the ones that could be subject of discussion.

L57-58: "the existence of two different low pressure areas is equivalent to the existence of two medicanes": do you mean that identifying two cyclones is a similar problem as the identification of two medicanes?

Yes. Provided that our algorithm searches the areas with high cyclonic potential and closed circulation, and isolates the points fulfilling the conditions to be a medicane center, it is technically the same thing searching for medicanes structures and searching for different cyclonic areas.

L244: is there a motivation for imposing - | VTL | > - | VTU |?

Yes, since the algorithm has a namelist-oriented conception, and it is prepared to track different types of cyclones, the inclusion of the 4th Hart condition enables the usage of the algorithm for tropical cyclones tracking. Although in medicanes literature this condition is usually not considered, the possibility to use it has been included for completeness and coherence. Since its checking can be deactivated in the namelist, its inclusion does not conflict with the physical argument that, for a medicane, the rate at which the geopotential height perturbation diminishes is not necessarily greater in the lower atmospheric layer than in the upper one.

L273: conversely, if the condition is valid, do you connect the positions at different i?

Let us explain this mathematical condition in detail. Given a medicane center at a certain position $M_t^c$ at a time step t, and if $DT_{max}$ equals one time step, we connect the found center with another one if in the previous or next time steps there exist a center at a distance lower than $D_{max}$ from $M_t^c$. In case of $DT_{max}$ higher than one time step, then we first check if there is a center separated one time step and at a distance lower

than $D_{max}$. If so, both are connected, and centers separated by more than one time step are not checked. If no center is found to be at a distance shorter than $D_{max}$ at one time step, then the centers at a distance of two time steps from t are checked to be at a spatial distance under $D_{max}$ from $M_t^c$, and so on until the number of time steps at which we check the centers meet the maximum $DT_{max}$. Please note that, by definition, when a center is found at a temporal distance of t-t' timesteps, the center is linked with that center and only that one. This ensures that each point is linked only once (if this is the case), and that we do not link centers too far in space and/or time.

Appendix A: "ZeroVortRadiusLowerLimit": occasionally Medicanes can be smaller than 80 km (see Miglietta et al., 2013)

This is precisely why ZeroVortRadiusLowerLimit is a namelist parameter. The possibility to adapt it to each type of structure and, even for a given structure like medicanes, to use the parameter value that each author considers appropriate is one of the main advantages of the conceived model. However, alghough most medicanes do shrink prior to its landfall, the zero vorticity radius is the mean distance to the line of zero vorticity, which usually exceeds the mark of 100 kilometers. Neither Miglietta et al. (2013) nor Tous and Romero (2013) (which Miglietta et al. (2013) cite for the medicanes radius) seem to provide a clear definition of how the measure the medicane radius. Thus, it is certainly difficult to know whether the medicane sizes they provide are comparable with the ones calculated with our proposed methodology (which also provides an inner radius, being the distance from the center to the point of maximum wind speed). In any case, thanks to the referee for pointing that out.

Figure E1: "Additionally, the green box covers the spatial area selected to run the algorithm on ERA5 data": is this for all Medicanes?

No. ERA5 reanalysis has only been used to provide a track of the Rolf medicane from a reanalysis global database, as presented in Figure 8. IC and BC for the WRF model run come from ERA-interim reanalysis data. Green box is the only one that does not

represent a domain for a model run, but a cropped window for using raw ECMWF reanalysis data as algorithm input, instead of using WRF output data.

**References**

Miglietta, M., Laviola, S., Malvaldi, A., Conte, D., Levizzani, V., and Price, C.: Analysis of tropical-like cyclones over the Mediterranean Sea through a combined modeling and satellite approach, Geophysical Research Letters, 40, 2400–2405, 2013.

Tous, M. and Romero, R.: Meteorological environments associated with medicane development, International Journal of Climatology, 33, 1–14, https://doi.org/10.1002/joc.3428, https://rmets.onlinelibrary.wiley.com/doi/abs/10.1002/joc.3428, 2013.

---

## Author Comment (AC2) · 15 Sep 2020

[1]EnriquePravia-Sarabia  [1]Juan  JoséGómez-Navarro  [1,2]PedroJiménez-Guerrero
[1]Juan PedroMontávez

[1]Physics of the Earth, Regional Campus of International Excellence "Campus Mare
Nostrum", University of Murcia, 30100 Murcia, Spain [2]Biomedical Research Institute of Murcia (IMIB-Arrixaca), 30120 Murcia, Spain Juan Pedro Montávez (montavez@um.es) 11

[Figure]

**TITAM (v1.0): Time Independent Tracking Algorithm for Medicanes. Reply to RC2.**

September 15, 2020

Referee comments: in blue.

Author responses: in black.

We welcome the feedback and appreciate all the referee suggestions and comments. While we do not agree with his/her general message , i.e. that he/she does not favour the publication of a purely algorithmic contribution, we agree that there is room for an improvement in the literature review regarding tropical cyclone tracking algorithms, and thus we have accordingly tried to adapt the manuscript to improve this caveat.

Related to the general comments:

The greatest contribution of this article are the criteria it uses to distinguish medicanes from typical extratropical cyclones. The use of the "cyclonic potential," and its motivation via connection to quasi-geostrophic theory is (to my knowledge) novel and a strength of the article.

We thank the reviewer for this comment.

These criteria are actually independent of the algorithm and software employed by the

authors which, as the title suggests, is a main focus of the article.

Even though the criteria are also valid for time-dependent methodologies, the main advantage is gained when used for time-independent, since it rapidly isolates the location of the medicane and, thus, it is not necessary to develop a method for searching the next center in the surroundings of the previous one.

Unfortunately, the software and algorithms advocated by the article are not new. Indeed, the algorithm is the same as the commonly used 2-step search that is described in greater detail by Bosler et.al. (2016), Ullrich Zarzycki (2017), Wernli Schweirtz (2006), and Zhao et. al. (2009), and each of these references succeed previous implementations of the same basic algorithm (e.g., Blender et.al. (1997), Hodges (1994), and Vitart et.al (1997) that are themselves successors of even earlier work. Even this list of references is, therefore, far from complete, and more specialized studies using this algorithm that are also relevant to this work include Hanley Caballero (2012), which (like the present work) addresses multiple circulations within the same larger system. None of the references in this review are cited by the current article, which is a significant omission, as it is not clear how (if at all) the present work distinguishes itself from them.

Perhaps this is a misunderstanding. Our algorithm is not exactly as those reviewed by the referee. Although undoubtedly our method takes ideas from previous ones, as we acknowledge in the literature review we provide (which of course can be and it is indeed improved in the new version), it tries to be more general and flexible. First, it is still based on an optimization problem for the search of the cyclone centre, but we introduce an additional level of complexity by allowing the possibility that the cyclone core is not necessarily the solution to the mathematical optimization point, but another one close to it that satisfies other physical conditions: optimizes a new function covering cyclonic potential, symmetry or warm core character. Thus, although it is not completely new in the fundamentals, we still claim its benefits when applied to the special problem of detecting and tracking medicanes in an automated way. Second, we have cited

all the works on which our method is based. Still, we greatly appreciate the list of additional references provided by the referee and have accordingly tried to improve the manuscript, more specifically the introduction section. Please note the changes associated with these comments in lines 64-94 of the changes version generated with the latexdiff tool.

As a consequence, I cannot recommend this article for publication.

This view is based on the two major concerns expressed below. Of course we do not agree with this recommendation, and we argue why in the response we give below to each detailed comment.

However, I would like to encourage the authors to reexamine their work from the context of their specific search criteria, which appear to be very successful at identifying medicanes. These methods could be applied to a larger dataset to examine the climatology of such storms, as in Zhao et. al (2009); the sensitivity of these climatologies could be examined with respect to different threshold values or criteria choices, as in Horn et. al. (2014). I also commend the authors for employing the cyclone "phase space" model as an analysis tool, and encourage them to continue to use it as this work matures.

Producing and analysing a medicanes climatology is, in fact, an interesting followup of our method, but we believe it is out of scope for this article. Clearly, an algorithm as the one presented here is a powerful tool for different studies. And this is precisely the reason why this method has been prepared for its publication separately. While we consider that the presented examples are enough to show its validity for medicanes, there are still some open questions about its implementation for tracking other types of cyclones. The namelist and the possibility of changing the different parameters provide a useful tool and opens a new discussion about the possibility of developing a single variable method for detection and tracking of different cyclones. Provided the great differences in scale and intensity between the tropical cyclones and the medicanes,

and given that the latter develop in baroclinic areas, the different tropical cyclones algorithms cited by the referee are worth a mention but still not valid for medicanes. Even though the time-independent methodology is not new in the tropical cyclones field, it is not so common in the medicanes tracking, as can be seen in the Table 1 (included upon request of referee 1). Apart from that, we have noticed that, if we are not wrong, no tropical cyclone tracking algorithm considers the center adjacent points as center candidates. This may lead to a misleading track if the found relative minimum does not fulfill the imposed conditions in multiple time steps. It is our understanding that provided the novel approach of the cyclonic potential measure (as the referee points out in the comments), along with the namelist-oriented implementation and the fact that some physical keys are thoroughly studied, such as the existence of a tilting and its relation with the slp minimum-warm core center deviation, or the consequences of using the slp minimum as the medicane center by using the Hart phase space, the manuscript contains sufficient information to be complete and valid on its own. Thus, the application to a larger dataset is beyond the scope of this contribution, although certainly it deserves being the main part of follow up studies. We are indeed working on applications aimed at improving medicanes process-understanding that we hope to send to revision within the next months. The objective of this contribution however is to follow the logic behind the development of the algorithm, while presenting, discussing and validating it, which in our view is very well aligned with the scope of a journal like GMD.

The focus in this ongoing work should not be on software. While writing software is undoubtedly where the authors spend much of their time and effort, this is the nature of modern science. The algorithms and code accompanying this article is not novel; it is simply one of many software packages that have been developed for similar applications in recent (and not-so recent) years. Instead, the application the authors have chosen is an excellent topic for additional study, particularly as resolution (both model and data set) increases to the point that medicanes are well-resolved.

We feel that this point of view is what has influenced the referee overall recommendation of rejecting our article. And we clearly do not agree. We consider that the focus we have given to the paper, aimed at describing the algorithm in a practical manner by means of examples and with parameter explanations and discussions, is best suited for a publication in a journal such as GMD, which is "dedicated to the publication and public discussion of the description, development, and evaluation of numerical models of the Earth system and its components". Our method, albeit not a completely new approach, is complex enough as to need a detailed description of its components, and that is why we have considered GMD as the appropriate journal to present it. We wish to re-emphasize here the fact that, contrary to the majority of former models, ours is open software, released and made easily accessible to the scientific community and prepared for an easy deployment, including libraries installation instructions. We think that accompanying the code with a GMD publication in which we describe the components and unravel the ideas that lie behind the algorithm is the best way to enable the development of new models, hence the perspective provided to the manuscript. Perhaps we have misunderstood the scope of this journal and its purpose, and hence we call for the Editor to help us to solve these two confronted viewpoints.

Now we proceed with the responses to the specific comments.

1. Figure 2(a): The legend label (SLP) is incorrect; the field shown is ∇2(SLP).

Thanks for noticing.

2. Line 298 reports that detected storms are shown in Figure 2(c). These are not visible in my .pdf copy. Also, it appears that there are many (56?) such detections – why? How does this number correspond to the number of time steps in the data set?

Thanks for pointing that out, they are in fact missing in the first version of the manuscript. Please find them in the new version. Also, provided the time-independent methodology, the number of points detected as candidates in each time step is not related in any way to the number of time steps. In fact, these 56 points are the ones in the

99.9 percentile (in a simulation with 200 x 280 grid points). This percentile, a namelist parameter, should be lowered in the presence of large cyclonic structures that could dominate the cyclonic potential field and eclipse other structures to be detected.

3. Line 305: What is gained by not using the SLP minimum as the location of the storm? In figure 4, the SLP minimum produces a clearly smoother track.

While in Figure 4 slp minimum track might seem smoother, note that the red points indicate that it does not fulfill the Hart conditions, and when analyzed in the Hart phase space, it could lead to wrong conclusions. Figures 3, 4 and 5 showcase the disadvantages of using the SLP minimum as the medicane center. Caption of Figure 4 has been rewritten for the sake of clarity.

4. Line 324: In what sense is the current method more "robust" than one that uses all the same criteria but chooses to define the location of the storm as the SLP minimum? This is one of many unquantifiable comments in the article that are better characterized as "sales" than scientific analysis.

Is more robust in the way that the SLP minimum is the point that is first checked (first candidate) but, as it does not always satisfies the conditions, alternative candidates are selected (please see Figure 4: green points are the timesteps in which the SLP minimum fulfills the criteria, red points the ones where another point must be selected as medicane center). In fact, SLP minimum is the one selected, but only if it is valid to be considered as medicane center. We do not intend to "sell" our approach, but to point out the advantages that can be easily demonstrated through these examples. In the light of the results in Figures 3, 4 and 5, it seems not pretentious to maintain that our method is more reliable than another using the SLP minimum (or any other relative maximum or minimum) alone, particularly if further studies in the Hart phase space are to be performed.

5. For a study intended to identify medicanes, the ability to distinguish a North Atlantic storm seems (as shown in Figure 7) seems irrelevant. A better example would be an

extratropical cyclone, associated with a digging trough, in one part of the Mediterranean basin and a separate system elsewhere in the region that contained a medicane, presuming such a situation can be found or simulated.

The Mediterranean situation the referee describes seems to be technically equivalent to the one shown with the North Atlantic storm. However, the one provided in the manuscript exhibits the additional ability of the proposed tracking method to isolate a medicane in the presence of a much larger and deeper low-pressure system, which was, a priori, prone to show large cyclonic potential values. Please note that in the presence of multiple large cyclonic systems, for example when using the method for global data in simultaneous presence of typhoons and hurricanes, the percentile parameter is probably something to take into account.

---

## Author Response (AR2)

**TITAM (v1.0): Time Independent Tracking Algorithm for Medicanes. Reply to minor revision.**

Enrique Pravia-Sarabia[1], Juan José Gómez-Navarro[1], Pedro Jiménez-Guerrero[1,2], and Juan Pedro Montávez[1]

[1]Physics of the Earth, Regional Campus of International Excellence "Campus Mare Nostrum", University of Murcia, 30100 Murcia, Spain
[2]Biomedical Research Institute of Murcia (IMIB-Arrixaca), 30120 Murcia, Spain

**Correspondence:** Juan Pedro Montávez (montavez@um.es)

Referee comments: in blue.

Author responses: in black.

We thank the referee for his/her useful and convenient suggestions. A major part of them have been included in the final manuscript version.

Now we proceed with the responses to the specific comments. Please note that the changes associated to these comments can be found in the changes version generated with the latexdiff tool.

L6: "the fact that the former ones appear in baroclinic environments makes them prone to maintain their warm-cored and symmetric structure for short time periods": really, also the limited extension of the Mediterranean basin contributes.

We agree, this is mentioned in the text (L46), but it has also been included in the abstract given the importance of this remark.

L10: based -> basing

We have accepted this change.

L93-95: "the existence of two different low pressure areas is equivalent to the existence of two medicanes": the sentence is still confusing, thus I suggest to change it, for example, following your previous answer, into "searching for two medicanes is technically the same as searching for two low pressure areas".

We have accepted this change.

L195: I think it would be clearer to change "in those cases where the medicane perturbation is largely modulated by orographic factors" into "in those cases where SLP perturbations occur due to orographic factors".

We have accepted this change.

L202: you should stress here that, due to the definition of vorticity, the threshold you selected depends on the grid spacing you used

We agree with this remark. We have included a short explanation on that in the final manuscript version (L204).

L243: higher or lower?

"Imposing a higher limit for the number of clusters prevents the inclusion of clusters not being real medicane candidates in large domains". It is an upper limit, it is the maximum number of medicane structures we allow to be present in the domain. This is useful "especially if the values selected for the previous filters were not tight enough". Thus, it prevents the points in

structures with low C values to be considered medicane candidates. We have changed "higher" to "upper" in the final manuscript version.

L284: the former being usually greater ...

We have accepted this change.

30 L354: really the value B = 10 m is appropriate for a radius of 500 km, thus one may question whether this value is appropriate for a much smaller radius. Figure 8 apparently supports this point.

Changing the B threshold in Figure 8 was associated to the coarser horizontal grid spacing of ERA5 data. Since the process of identifying the medicane center, as well as the medicane boundaries and the points in which the vorticity drops to zero, is not as precise as for finer resolutions, we introduce a higher degree of variability (increasing B threshold) so that we do not

35 miss points due to the grid spacing. For finer horizontal grid spacings, the value should not change even for larger medicane structures, since they probably still maintain the axisymmetrical structure when considering their complete extent. However, the suitable B value to use is undoubtedly an interesting discussion that should be addressed on each particular case, and that is the reason why its value is a namelist parameter that can be adjusted to user convenience.

Figure 5 caption: it is not clear what is the latitude of the zonal cross section: that passing through the SLP minimum?

40 through the position of the cyclone? or another one?

It is a zonal cross section at the latitude of the medicane center found by the tracking algorithm. We have further clarified this in caption of Figure 5, which now includes a detailed description of the latitude along which zonal cross section is shown.

L383: what do you mean here with "instability"?

Contrary to the TC, where a stable vertical structure is established and maintained, the medicanes tilting make them prone

45 to lose their structure and rapidly weaken. We have rephrased this part (L384-385) to make it more clear and avoid using the instability concept in such an ambiguous way.

L397 and Figure 7 caption: large plus symbols, respectively ... (I do not see any red plus in the Mediterranean)

It is there, but is difficult to see as it overlaps with the green plus symbol. It is visible with the zoom box presented at the bottom left corner of the right plot in Figure 7.

50 L422: I tried hard to understand why the track of CY (reaching Crete in Fig. 9) stopped so early in Fig. 10. It seems that the explanation is provided at the end ("the algorithm does not follow the SYNOPTIC low since it does not satisfy other conditions such as the symmetry"): is my interpretation right? Consider to anticipate the explanation earlier.

It is right. We have anticipated the explanation, and included it also in the caption of Figure 10 for the sake of clarity.

L453: the selected examples ...

55 We have accepted this change.

Comment on the previous answer Neither Miglietta et al. (2013) nor Tous and Romero (2013) (which Miglietta et al. (2013) cite for the medicanes radius) seem to provide a clear definition of how the measure the medicane radius. Miglietta et al. (2013) state (final sentence at page 2401) that "After several trials and errors, here the radius is chosen considering the extension of the warm core anomaly at 600 hPa, as implicitly suggested by Hart".

60     We agree. However, it is still not clear for us what is the form in which the 600 hPa anomaly leads to a radius measure. From Hart (2003), they seem to use a vertical cut and analyze the anomaly in one direction. This definition is suitable when there is a very strong axisymmetric structure (such as in TC), but may lead to unexpected results in case the medicane associated anomaly is not perfectly axisymmetric. ¿What radius should be chosen in that case, a mean along various directions, the maximum, the minimum, other? This is what we mean when we explain that no definition is provided since, to our knowledge, there is not an

65   explicit measure that can be reproduced in all medicane cases.

[revised manuscript text omitted]